# Genomic insights into the host specific adaptation of the *Pneumocystis* genus

Ousmane H. Cissé [1,17✉], Liang Ma [1,17✉], John P. Dekker[2,3], Pavel P. Khil[2,3], Jung-Ho Youn[3], Jason M. Brenchley[4], Robert Blair [5], Bapi Pahar [5], Magali Chabé [6], Koen K. A. Van Rompay [7], Rebekah Keesler[7], Antti Sukura[8], Vanessa Hirsch [9], Geetha Kutty[1], Yueqin Liu[1], Li Peng[10], Jie Chen [10], Jun Song [11], Christiane Weissenbacher-Lang[12], Jie Xu [11], Nathan S. Upham[13], Jason E. Stajich [14], Christina A. Cuomo [15], Melanie T. Cushion [16] & Joseph A. Kovacs [1✉]

*Pneumocystis jirovecii*, the fungal agent of human *Pneumocystis* pneumonia, is closely related to macaque *Pneumocystis*. Little is known about other *Pneumocystis* species in distantly related mammals, none of which are capable of establishing infection in humans. The molecular basis of host specificity in *Pneumocystis* remains unknown as experiments are limited due to an inability to culture any species in vitro. To explore *Pneumocystis* evolutionary adaptations, we have sequenced the genomes of species infecting macaques, rabbits, dogs and rats and compared them to available genomes of species infecting humans, mice and rats. Complete whole genome sequence data enables analysis and robust phylogeny, identification of important genetic features of the host adaptation, and estimation of speciation timing relative to the rise of their mammalian hosts. Our data reveals insights into the evolution of *P. jirovecii*, the sole member of the genus able to infect humans.

[1] Critical Care Medicine Department, NIH Clinical Center, National Institutes of Health (NIH), Bethesda, MD, USA. [2] Bacterial Pathogenesis and Antimicrobial Resistance Unit, National Institute of Allergy and Infectious Diseases (NIAID), NIH, Bethesda, MD, USA. [3] Department of Laboratory Medicine, NIH Clinical Center, National Institutes of Health, Bethesda, MD, USA. [4] Laboratory of Viral Diseases, NIAID, NIH, Bethesda, MD, USA. [5] Tulane National Primate Research Center, Tulane University, New Orleans, LA, USA. [6] Univ. Lille, CNRS, Inserm, CHU Lille, Institut Pasteur de Lille, U1019-UMR 9017-CIIL-Centre d'Infection et d'Immunité de Lille, Lille, France. [7] California National Primate Research Center, University of California, Davis, CA, USA. [8] Department of Veterinary Pathology, Faculty of Veterinary Medicine, University of Helsinki, Helsinki, Finland. [9] Laboratory of Molecular Microbiology, NIAID, NIH, Bethesda, MD, USA. [10] Department of Respiratory and Critical Care Medicine, the First Affiliated Hospital of Chongqing Medical University, Chongqing, China. [11] Center for Advanced Models for Translational Sciences and Therapeutics, University of Michigan Medical Center, University of Michigan Medical School, Ann Arbor, MI, USA. [12] Institute of Pathology, Department of Pathobiology, University of Veterinary Medicine Vienna, Vienna, Austria. [13] Arizona State University, School of Life Sciences, Tempe, ARI, USA. [14] Department of Microbiology and Plant Pathology and Institute for Integrative Genome Biology, University of California, Riverside, Riverside-California, Riverside, CA, USA. [15] Broad Institute of Harvard and Massachusetts Institute of Technology, Cambridge, MA, USA. [16] Department of Internal Medicine, College of Medicine, University of Cincinnati, Cincinnati, OH, USA. [17]These authors contributed equally: Ousmane H. Cissé, Liang Ma. ✉email: ousmane.cisse@nih.gov; mal3@nih.gov; jkovacs@nih.gov

The evolutionary history of *Pneumocystis jirovecii*, a fungus that causes life-threatening pneumonia in immunosuppressed patients such as those with HIV infection, has been poorly defined. *P. jirovecii* is derived from a much broader group of host-specific parasites that infect all mammals studied to date. Until recently, *P. carinii* and *P. murina* (which infect rats and mice, respectively) were the only other species in this genus for which biological specimens suitable for whole-genome sequencing were readily available. Cross-species inoculation studies of *P. jirovecii* and *P. carinii* have found that they can only infect humans and rats, respectively[1,2]. Further, rats are the only mammals known to be coinfected by at least two distinct *Pneumocystis* species (*P. carinii* and *P. wakefieldiae*)[3]. Within the *Pneumocystis* genus, *P. jirovecii* is the only species able to infect and reproduce in humans, although the molecular mechanisms of its host adaptation remain elusive.

Previous efforts to reconstruct the evolutionary history of *Pneumocystis* have estimated the origins of the genus at a minimum of 100 million years ago (mya)[4]. Using a partial transcriptome of *P. sp. macacae* (hereafter referred to as *P. macacae*), the *Pneumocystis* species that infects macaques, we recently estimated that *P. jirovecii* diverged from the common ancestor of *P. macacae* around ~62 mya[5], which substantially precedes the human-macaque split of ~20 mya[6]. Population bottlenecks in *P. jirovecii* and *P. carinii* at 400,000 and 16,000 years ago, respectively[5], are also not concordant with population expansions in modern humans (~200,000 years ago[7]) and rats (~10,000 years ago[8]), which suggests a decoupled coevolution between *Pneumocystis* and their hosts. Thus, *Pneumocystis* species may not be strictly coevolving with their mammalian hosts as suggested by ribosomal RNA-based maximum phylogenies[9]. A molecular clock has not been tested in any of these phylogenies. A strict coevolution hypothesis has been further challenged by evidence suggesting a relaxation of the host specificity in *Pneumocystis* infecting rodents[10,11]. However, the accuracy of speciation times is limited without the complete genomes of additional species including that of *P. macacae*, the closest living sister species to *P. jirovecii* identified to date.

The absence of long-term in vitro culture methods or animal models for most *Pneumocystis* species has precluded obtaining sufficient DNA for full genome sequencing and has hindered investigation of the *Pneumocystis* genus. To date, only the genomes of human *P. jirovecii*[12,13], rat *P. carinii*[13,14], and mouse *P. murina*[13], are available. These data have provided important insights into the evolution of this genus, including a substantial genome reduction[12,13], the presence of intron-rich genes possibly contributing to transcriptome complexity, and an expansion of a highly polymorphic major surface glycoprotein (*msg*) gene superfamily[13], some of which are important for immune evasion. However, the lack of whole-genome sequences for many species of this genus (particularly the closely related *P. macacae*) has severely constrained the understanding of the implications of these genome features in *Pneumocystis* evolution and adaptation to hosts.

To explore the evolutionary history of the *Pneumocystis* genus, and investigate *P. jirovecii* genetic factors that support its adaptation to humans, we sequenced 2–6 specimens of four additional species: those that infect macaques (*P. macacae*), rabbits (*P. oryctolagi*), dogs (*P. canis*), and rats (*P. wakefieldiae*).

## Results

### Direct sequencing of *Pneumocystis*-host mixed samples.

We sequenced the genomes of *Pneumocystis* species from infected macaques, rabbits, dogs, and rats (see Methods and Supplementary Methods). Specimens originated from immunosuppressed

animals as a consequence of simian immunodeficiency virus infection in macaques, corticosteroid treatment (rabbits and rats), immunodeficient knockout (rabbits) or possible congenital immunodeficiencies (dogs). For each species, we sequenced multiple samples from 2–6 animals (Supplementary Tables 1 and 2). These data were used to assemble one nearly full-length genome assembly for each species except *P. canis* for which we recovered two nearly full-length assemblies and an additional partial assembly from two separate samples (denoted as A, Ck1, and Ck2). Post assembly mapping revealed a negligible amount of genetic variability among samples, for example the average genome-wide single-nucleotide polymorphism (SNP) diversity among six *P. macacae* isolates excluding highly polymorphic regions such as Msg genes is ~0.1%. The genome of *P. macacae* was sequenced using Oxford Nanopore long reads and Illumina short read sequences, whereas the other *Pneumocystis* were sequenced only with Illumina (Supplementary Tables 2 and 3). The genome assemblies range from 7.3 Mb in *P. wakefieldiae* to 8.2 Mb in *P. macacae*. The *P. macacae* and *P. wakefieldiae* genome assemblies consist of 16 and 17 scaffolds, respectively, both of which are highly contiguous and approach the chromosomal level based on similarities with published karyotypes[3,15] and/or the presence of *Pneumocystis* telomere repeats[16] at the scaffold ends (Supplementary Table 3). The genome assemblies of *P. oryctolagi* and *P. canis* (assemblies A, Ck1, and Ck2) are less contiguous with 38, 33, 78, and 315 scaffolds, respectively. All these assemblies except for the partial assembly of *P. canis* Ck2 have very similar total sizes (7.3–8.2 Mb) comparable to previously sequenced genomes of *P. jirovecii*, *P. carinii*, and *P. murina*, all of which are at or near chromosomal level with a size of 7.4–8.3 Mb (Supplementary Table 3). The genome assemblies are all AT-rich (~71%) and ~3% encode DNA transposons and retrotransposons (Supplementary Table 3). We also assembled complete mitochondrial genomes from all species in this study, which are similar in size (21.2–24.5 kilobases) to published rodent *Pneumocystis* mitogenomes (24.6–26.1 kb)[17] but smaller than that of *P. jirovecii* (~35 kb)[17] (Supplementary Table 3). *P. macacae* has a circular mitogenome similar to *P. jirovecii*[17] whereas all other sequenced species have linear mitogenomes.

### Genomic differences among *Pneumocystis* species.

To assess the extent of genome structure variations among species, we generated whole-genome alignment of all representative genome assemblies. We found high levels of interspecies rearrangements ranging from 10 breakpoints between *P. wakefieldiae* and *P. murina* to 142 between *P. jirovecii* and *P. oryctolagi* (Fig. 1; Supplementary Table 4). The vast majority of chromosomal rearrangements were inversions, which, for example accounted for 23 out of 29 breakpoints between *P. jirovecii* and *P. macacae* (Supplementary Table 4). Analysis of aligned raw Nanopore and/ or Illumina reads back to the assemblies show no evidence of incorrect contig joins around rearrangement breakpoints. There are clearly fewer rearrangements among rodent *Pneumocystis* species (*P. wakefieldiae*, *P. carinii*, and *P. murina*) than among all other species (Fig. 1; Supplementary Table 4), which is likely due to the younger evolutionary ages of rodent *Pneumocystis* (Fig. 2a and c). These rearrangements could have caused incompatibilities between species, thus preventing gene flow for species that infect the same host.

Comparison of pairwise whole-genome alignment identities between species indicates a substantial nucleotide divergence: 14% dissimilarity in aligned regions between *P. jirovecii* and *P. macacae*; 21% between *P. jirovecii* and *P. oryctolagi*; 22% between *P. jirovecii* and *P. canis* Ck1; 15% between *P. wakefieldiae*

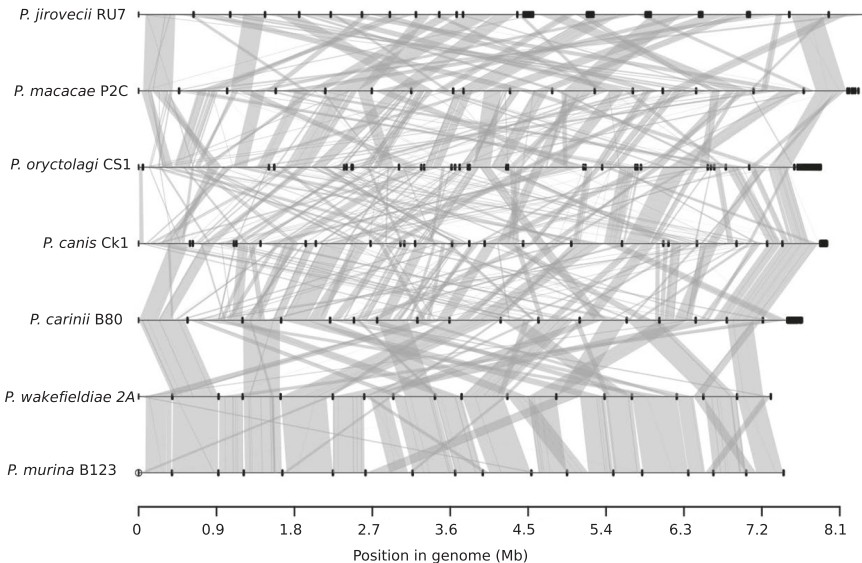

**Fig. 1 Whole-genome structure and synteny among *Pneumocystis* species.** Species names and their genome assembly identifiers are shown on the left. Horizontal black lines on the right represent sequences of all scaffolds for each genome laid end-to-end, with their nucleotide positions indicated at the bottom. Dark thick squares represent short scaffolds. Syntenic regions between genomes are linked with vertical gray lines.

and *P. carinii*; and 12% between *P. wakefieldiae* and *P. murina* (Supplementary Table 5).

To understand the relationship between the intraspecies and interspecies genetic diversity of *Pneumocystis*, we generated additional four *P. jirovecii* and five *P. macacae* genome assemblies from low sequence coverage samples. All data are expressed as the mean ± standard deviation. The pairwise intraspecies genome divergences among *P. jirovecii* genome assemblies ($0.3 ± 0.2\%$, $n = 8$) are significantly lower than those obtained when comparing them to *P. macacae* assemblies ($16.1 ± 0.2\%$, $n = 5$) (two sample $t$-test, $p$-value $= 1.4 × 10^{-14}$) or to other *Pneumocystis* species ($21.6 ± 0.9\%$, $n = 7$) ($p = 2.9 × 10^{-10}$). Similarly, mean divergence among *P. macacae* genome assemblies ($0.8 ± 0.3\%$, $n = 5$) is lower than divergence when they are compared to *P. jirovecii* assemblies ($15.6 ± 0.2\%$, $n = 8$) ($p = 1.3 × 10^{-10}$) or other *Pneumocystis* species genome assemblies ($21.8 ± 0.9\%$) ($p = 7.2 × 10^{-11}$). The results indicate that interspecies divergence exceeds intraspecies divergence, which is consistent with a complete species separation.

**Speciation history of the *Pneumocystis* genus.** These new complete genome data enabled us to examine the relationships between different *Pneumocystis* species and to estimate the timing of speciation events that led to the extant species. We inferred a strongly supported phylogeny of *Pneumocystis* species rooted with outgroups from distantly related fungal subphyla. Our phylogenomic analysis of 106 single-copy orthologs inferred from all assemblies including the fragmented Ck2 strongly supports monophyly of *Pneumocystis* species (100% Maximum likelihood bootstrap values; Fig. 2a), Bayesian posterior probabilities (>0.95; Supplementary Fig. 1), and highly significant support from the Shimodaira–Hasegawa test[18] ($p < 0.001$; see Methods). An identical phylogeny was recovered using mitochondrial genome data from 33 specimens representing 7 *Pneumocystis* (Supplementary Fig. 2). However, we identified unexpected placements of *P. wakefieldiae*, *P. oryctolagi*, and *P. canis*. First, *P. wakefieldiae* appears as a sister species of *P. murina* instead of *P. carinii* (which also infects rats) (Fig. 2b). This observation is supported by the higher similarity in genome size (Supplementary Table 3), lower sequence divergence (Supplementary Table 4), higher

genome synteny (Fig. 1; Supplementary Table 5) and higher frequencies of supporting genes (0.64 in 1,718 nuclear gene trees examined; Methods) between *P. wakefieldiae* and *P. murina* than between *P. wakefieldiae* and *P. carinii*. These relationships contradict the previous phylogenetic placement of *P. wakefieldiae* as an outgroup of the *P. carinii*/*P. murina* clade[9] or a sister species of *P. carinii*[19] based on analysis of mitochondrial large and small subunit rRNA genes (mtLSU and mtSSU). Our phylogeny also opposes the prevailing hypothesis for dynamics of host specificity and coevolution within the *Pneumocystis* genus, that is, *P. wakefieldiae* shares with *P. carinii* the same host species (*Rattus norvegicus*) and thus is expected to be more related to *P. carinii* than to *P. murina*.

Similarly, *P. oryctolagi* would be expected to be phylogenetically closer to rodent *Pneumocystis* than to primate *Pneumocystis*, consistent with the closer phylogenetic relationships of rabbits and rodents to each other than to primates[20] (Fig. 2a, b). In contrast, *P. oryctolagi* and *P. canis* are more closely related to primate *Pneumocystis* (*P. jirovecii* and *P. macacae*) than rodent *Pneumocystis* (Fig. 2a; Supplementary Figs. 1 and 2; 100% of tree level support in 1718 nuclear genes). The phylogenetic discrepancy between *P. oryctolagi* and its host (rabbit) suggests that host switching may have occurred in their distant history.

From whole-genome Bayesian phylogenetic estimates (see Methods), the common ancestor of all extant species of the genus emerged around 140 mya (confidence intervals: 180–101 mya; Fig. 2c; Supplementary Fig. 1), with a separation of *Pneumocystis* and *Schizosaccharomyces* genera around 512 mya (CI: 822–203 mya), which is consistent with independent estimates of the origin of Taphrinomycota crown group at 530 mya[21]. The *Pneumocystis* genus thereafter divided into two main clades, P1 consisting of *P. jirovecii*, *P. macacae*, *P. oryctolagi*, and *P. canis*, and P2 consisting of species infecting rodents (*P. carinii*, *P. wakefieldiae*, and *P. murina*) (Fig. 2b). Subsequent to the divergence of P1/P2, the clade P1 diversified through a series of speciation events leading either to primate or carnivore species whereas P2 remained localized in rodents. We also found that the divergence time of *Pneumocystis* in the clade P1 predates that of their hosts, that is, the crown of rodent-rabbit-primate

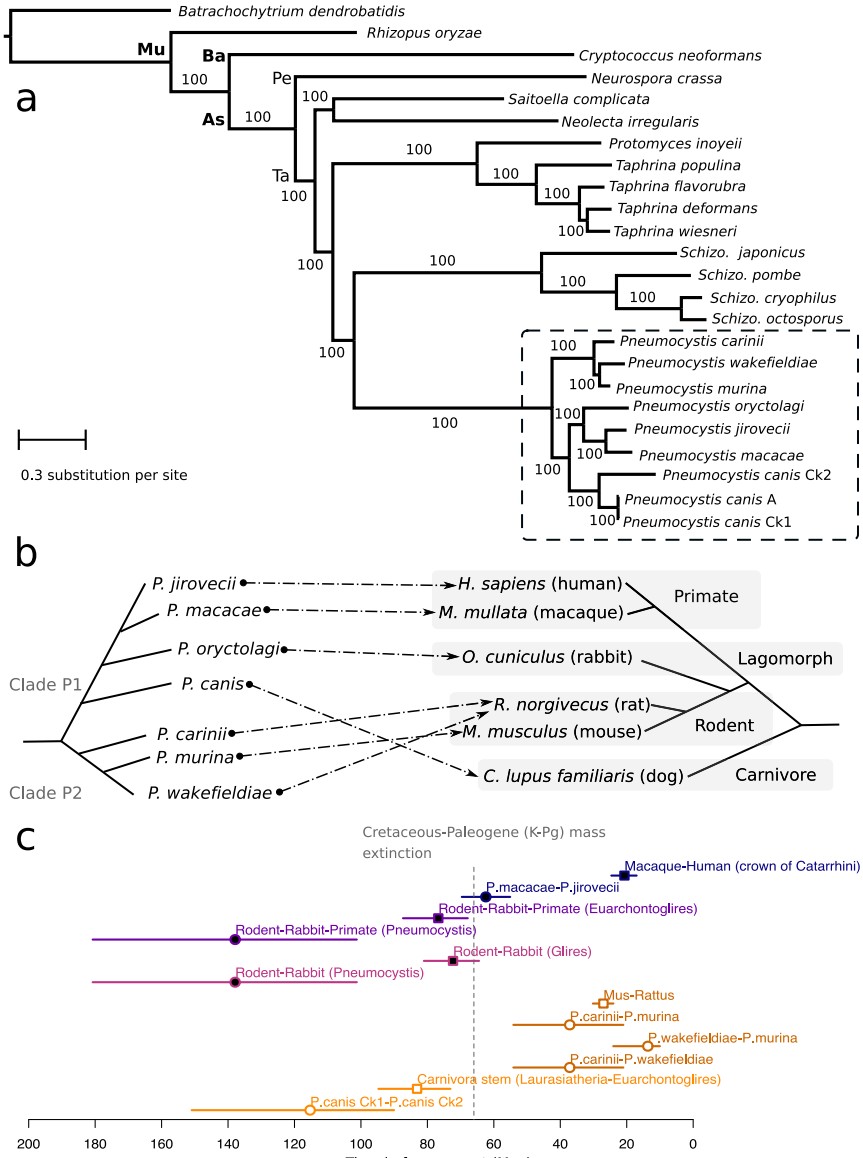

**Fig. 2 Phylogeny and divergence times of *Pneumocystis* species. a** Maximum likelihood phylogeny constructed using 106 single-copy genes based on 1000 replicates from 24 annotated fungal genome assemblies including nine from *Pneumocystis* (highlighted with a dashed box). Only one assembly is shown for each species except there are three for *P. canis* (assemblies Ck1, Ck2, and A). Bootstrap support (%) is presented on the branches. The fungal major phylogenetic phyla and subphyla are represented by their initials: *As* (Ascomycota), *Ba* (Basidiomycota), *Pe* (Pezizomycotina), *Mu* (Mucoromycota), and *Ta* (Taphrinomycotina). **b** Schematic representation of species phylogeny and association between *Pneumocystis* species and their respective mammalian hosts. The dashed arrows represent the specific parasite-host relationships. **c** Divergence times of *Pneumocystis* species and mammals (*n* = 12 taxonomic clades analyzed). Divergence time medians are represented as squares for hosts and as circles for *Pneumocystis*, and the horizontal lines represent the 95% confidence interval (CI) error bars, which are color-coded the same for each *Pneumocystis* and its host. Closed elements represent nodes that are different in term of divergence times (nonoverlapping confidence intervals) whereas open elements represent nodes with overlapping confidence intervals. Catarrhini, taxonomic category (parvorder) including humans, great apes, gibbons, and Old-World monkeys. Euarchontoglires, superorder of mammals including rodents, lagomorphs, treeshrews, colugos, and primates. Glires, taxonomic clade consisting of rodents and lagomorphs. Laurasiatheria, taxonomic clade of placental mammals that includes shrews, whales, bats, and carnivorans. Mya, million years ago. K-Pg, Cretaceous-Paleogene. The dotted vertical line representing the K-Pg mass extinction event at 66 mya is included for context only.

*Pneumocystis* is clearly more ancient than the corresponding superorder of mammals (Euarchontoglires) (Fig. 2c). The pattern in clade P2 is different as the divergence time estimates overlap with those of their hosts (Fig. 2c). On the basis of coalescent estimates, *P. jirovecii* began to split from *P. macacae* at ~62 mya (CI: 69–55 mya), which extended through the Cretaceous-Paleogene mass extinction event at 66 mya, but substantially

predates the crown Catarrhini (human-macaque ancestor) of ~20 mya (CI: 24–17 mya; Fig. 2c; Supplementary Fig. 1).

**High levels of population differentiation among *Pneumocystis* genomes support reproductive isolation**. To understand the genomic divergence landscape of *Pneumocystis* populations, we performed genome-wide differentiation tests ($F_{ST}$, relative

population divergence) and nucleotide diversity ($\pi$). These analyses used 32 genomic datasets, including 26 publicly available datasets in GenBank for *P. jirovecii*, *P. carinii* and *P. murina* and six datasets generated in this study for other four *Pneumocystis* species (Supplementary Note 1; Supplementary Table 2). We used a trained version LAST[22] to account for interspecies divergence during read mapping and ANGSD[23] to derive genotype likelihoods instead of genotypes. Since ANGSD's $F_{ST}$ requires outgroups, we analyzed interspecies divergence between *P. jirovecii*, *P. macacae*, and *P. oryctolagi* populations using a sliding window approach of 5-kb and *P. carinii* as an outgroup species (*n* samples = 59). *P. murina* genomic divergence relative to *P. carinii* and *P. wakefieldiae* populations was estimated similarly using *P. jirovecii* as an outgroup species (*n* = 47). We found high levels of population differentiation among *Pneumocystis* specimens; 71.9% of the *P. jirovecii* genome had a Fixation index ($F_{ST}$) > 0.8 compared to the closest species, *P. macacae*, while 90.2% of the genome had a $F_{ST}$ > 0.8 compared to the extant species *P. oryctolagi* (Supplementary Fig. 3). Similarly, 86.3% and 93.7% of the *P. murina* genome had a $F_{ST}$ > 0.8 compared to *P. carinii* and *P. wakefieldiae*, respectively (Supplementary Note 1).

**Analyzing historical hybridization in *Pneumocystis* genus.** Topology-based maximum likelihood analysis of 1718 gene trees using PhyloNet[24] found no evidence of gene flow among species of clade P1 (*P. jirovecii*, *P. macacae*, *P. oryctolagi*, and *P. canis*) (Supplementary Fig. 4), which indicates that these lineages were reproductively isolated throughout their evolutionary history, consistent with their isoenzyme diversity[25]. In contrast, we found strong evidence of ancient hybridization in clade P2, possibly between *P. carinii* and *P. wakefieldiae* (Supplementary Fig. 4), which may then have contributed to the formation of the *P. murina* lineage. We hypothesize that *P. murina* might have originated as a hybrid between ancestors of *P. carinii* and *P. wakefieldiae* in rats, and subsequently shifted to mice, possibly owing to the geographic proximity of ancestral rodent populations (for example in Southern Asia[26]), which is consistent with the fact that ecological fitting is a major determinant of host switch[27]. The presumed physiological, cellular and/or immunological similarities among closely related rodent species might also have helped the same *Pneumocystis* species colonizing multiple closely related rodent species[10,27].

**Gene families and metabolic pathways linked to host specificity.** The predicted protein-coding gene numbers are similar across *Pneumocystis* genomes and range from 3211 in *P. wakefieldiae* to 3476 in *P. canis* strain Ck1 (Supplementary Table 3). Nearly all predicted protein-coding genes in *P. macacae* (96% of 3471) and *P. wakefieldiae* (99% of 3221) genomes have RNA-Seq support. Gene models present a complex architecture with ~6 exons per gene on average. High representation of core eukaryotic genes in *P. macacae*, *P. oryctolagi*, *P. canis* and *P. wakefieldiae* provides evidence that these genomes are nearly complete and comparable in completeness to *P. jirovecii*, *P. murina*, and *P. carinii* genomes: 86.2–93.4% of conserved genes are detectable in all annotated genome assemblies (Supplementary Table 3).

Examination of orthologous genes reveals that ~3100 orthologous clusters had representative genes from all nine analyzed genome assemblies from seven *Pneumocystis* species (Supplementary Table 3). We found a small number of unique genes in each *Pneumocystis* species ranging from 25 in *P. wakefieldiae* to 204 in *P. oryctolagi* (Supplementary Table 3). Unique genes in most species encode for phylogenetically unrelated proteins with unknown function. A striking exception is observed in *P. macacae* in which nearly all unique proteins are part of an

undescribed large protein family (*n* = 190). The members of this family are enriched in arginine and glycine amino-acid residues (denoted RG proteins) (Supplementary Fig. 5a) and have no similarities with transposable elements. While RG motifs are often found in eukaryotic RNA-binding proteins[28], *P. macacae* RGs do not possess an RNA-binding domain (Pfam domains PF00076, PF08675, PF05670, PF00035), suggesting a different role. In addition, *P. macacae* RGs lack functional annotation except for two proteins that encode a Dolichol-phosphate mannosyltransferase domain (PF08285) and a leucine zipper domain (PF10259), respectively. Of the 190 RGs, 134 have RNA-Seq based gene expression support, including five among the top highly expressed genes (Supplementary Fig. 5b). Nearly half of RGs are located at subtelomeric regions, often found in close proximity to *msg* genes (Supplementary Data 1). RG proteins can be grouped in three main clusters (based on OrthoFinder clustering; Methods), have a reticulate phylogeny (Supplementary Fig. 5c) and a mosaic gene structure (Supplementary Fig. 5d), which suggest frequent gene conversion events.

To investigate the gene loss patterns in sequenced genomes, we compared *Pneumocystis* gene catalogs to those of related Taphrinomycotina fungi. We found that all sequenced *Pneumocystis* species have lost ~40% of gene families present in other Taphrinomycotina (Supplementary Fig. 6), and that the metabolic pathways are also very similar among *Pneumocystis* species with a few minor (possibly stochastic) differences (Supplementary Note 2). This strongly suggests that *Pneumocystis* ancestry experienced massive gene losses that occurred before the genus diversification.

To investigate changes in gene content that might explain interspecies differences among the seven *Pneumocystis* species, we searched for expansions or contractions in functionally classified gene sets. We identified Pfam domains with significantly uneven distribution among genomes (Wilcoxon signed-rank test $p < 0.05$). Domains associated with Msg proteins are enriched in *P. jirovecii* and, to a lesser extent in *P. macacae* compared to other species (Fig. 3a). Domains associated with peptidases (M16) are enriched in *P. carinii*, *P. murina*, and *P. wakefieldiae*. S8 peptidase family (kexin) is expanded in *P. carinii* with 13 copies[13] whereas all other species have one or three copies (Fig. 3a; Supplementary Fig. 7). Although kexin is localized in other fungi to the Golgi apparatus, and in *Pneumocystis* is believed to be involved in the processing of Msg proteins, the expanded copies are predicted to be GPI-anchored proteins, and appear to localize to the cell surface; their function is unknown[29]. We found that *P. carinii* kexin genes evolved under strong positive selection ($p = 0.008$) whereas *P. wakefieldiae* kexin genes did not ($p = 0.159$).

Phylogenetic analysis of CFEM (common in fungal extracellular membrane) protein domains, which are important for the acquisition of vital compounds in fungi[30], suggest that these domains were likely already present in the last common ancestor of *Pneumocystis* and were vertically transmitted (Supplementary Fig. 8).

To investigate changes in enzyme gene content that might account for interspecies differences among *Pneumocystis* species, we searched for enzymes that show clear differences among species, which are represented by Enzyme Commission numbers (ECs) (Fig. 3b). We found 34 ECs, which include 14 that are highly conserved in *P. jirovecii* but have a patchy distribution in other members of clade P1 (*P. macacae*, *P. canis*, and *P. oryctolagi*) and are lost in clade P2 (*P. carinii*, *P. murina*, and *P. wakefieldiae*). Most of these 14 ECs are assigned to the biosynthesis of antibiotics or secondary metabolites and vitamin B6 metabolism according to KEGG pathways. The latter pathway seems only functional in P2 clade (Supplementary Note 2).

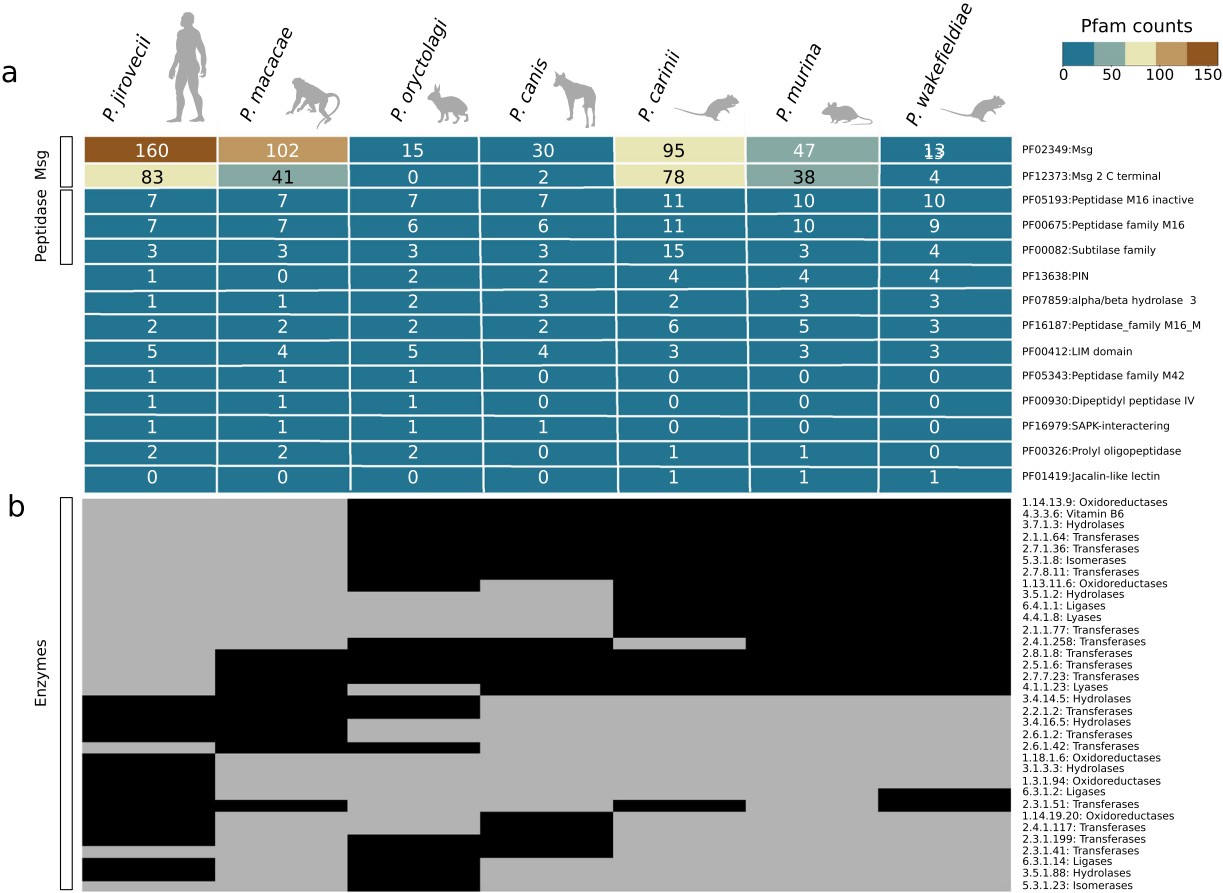

**Fig. 3 Distribution of protein families among *Pneumocystis* species. a** Heatmap of Pfam protein domains with significant differences (Wilcoxon signed-rank test, *p* < 0.05) are included if the domain appears at least once in the following comparisons: primate *Pneumocystis* (*P. jirovecii* and *P. macacae*) versus other *Pneumocystis*, clade P1 (*P. jirovecii, P. macacae, P. oryctolagi*, and *P. canis* Ck1) versus clade P2 (*P. carinii, P. murina*, and *P. wakefieldiae*), primate *Pneumocystis* versus clade P2. The number of proteins containing each domain is indicated within each cell for each species. The heatmap is colored according to a score, as indicated by the key at the upper right corner. **b** Heatmap of distribution of enzymes (represented by Enzyme Commission numbers and their KEGG functional categories), with their presence and absence indicated by black and grey colored cells, respectively. Animal icons were obtained from http://phylopic.org under creative commons licenses https://creativecommons.org/licenses/by/3.0/: mouse (Anthony Caravaggi; license CC BY-NC-SA 3.0); dog (Sam Fraser-Smith and vectorized by T. Michael Keesey; license CC BY 3.0), rabbit (by Anthony Caravaggi; license CC BY-NC-SA 3.0), and rat (by Rebecca Groom; license CC BY-NC-SA 3.0). Icons original black color background were modified to light gray.

**Intron evolution**. We analyzed 1211 one-to-one gene orthologs shared by all sequenced *Pneumocystis* and other Taphrinomycotina fungi (Supplementary Fig. 9a). A total of 9080 homologous sites within 1211 alignments were identified (Supplementary Fig. 9b). While intron densities are similar among *Pneumocystis* species (ranging from 4842 in *P. macacae* to 5289 in *P. murina*), they are markedly more elevated compared to related Taphrinomycotina, including *Neolecta irregularis* ($n = 4202$ introns), *Schizosaccharomyces pombe* ($n = 862$), and *Taphrina deformans* ($n = 639$) (Supplementary Fig. 11b). Predictions of ancestral intron densities show that the *Pneumocystis* common ancestor had at least 5341 introns, of which 37% were not found in other Taphrinomycotina (Supplementary Fig. 9c). This is in contrast to other fungi; ~26% of *Neolecta* introns were independently acquired whereas *S. pombe* and *T. deformans* genomes have experienced intron losses, which is consistent with published studies[31,32]. These results suggest that the emergence of *Pneumocystis* genus was preceded by gain of introns.

**Positive selection footprints in *P. jirovecii* genes**. We tested the hypothesis that *P. jirovecii* has adapted specifically to humans after its separation with *P. macacae*, and that there will be footprints of

directional selection in the genome that point to the molecular mechanisms of this adaptation. To infer *P. jirovecii*-specific adaptive changes, we compared the *P. jirovecii* one-to-one orthologs to those of *P. macacae* and *P. oryctolagi* using the branch-site likelihood ratio test[33]. Positive selection was identified as an accelerated nonsynonymous substitution rate. The test identified 244 genes (out of 2466) with a signature of positive selection in the human pathogen *P. jirovecii* alone (Bonferroni corrected *p*-value < 0.05; Supplementary Data 2). Gene Ontology enrichment analysis of these genes with accelerate rates identified significant enrichment for the biological process "cellular response to stress" (adjusted using Benjamini–Hochberg *p*-value = $1.9 \times 10^{-6}$) and the molecular function "potassium channel regulator activity" ($p = 2.8 \times 10^{-10}$). Among the 244 genes, 197 are conserved in all *Pneumocystis* genomes available whereas 47 are absent in clade P2 only (*P. carinii, P. murina*, and *P. wakefieldiae*; Fig. 2b). While the latter set of genes encode proteins of unknown function, analysis of Pfam domains shows a significant enrichment in the biological process "nucleoside phosphate biosynthetic" process ($p = 9.9 \times 10^{-5}$) and the molecular function "carbon–nitrogen lyase activity" ($p = 2.8 \times 10^{-10}$). Further investigations will be required to determine the precise functions of these genes.

**Subtelomeric gene families**. Until recently, the only in-depth data on the subtelomeric gene families in *Pneumocystis* have come from the *P. jirovecii*, *P. carinii*, and *P. murina* genomes[13,34]. These genes, including *msg* and kexin, are believed to be important for antigenic variation, and are well represented in the assemblies of *P. macacae*, *P. oryctolagi*, *P. canis*, and *P. wakefieldiae*.

*P. macacae* subtelomeres encode numerous arrays of Msg and RG proteins (Supplementary data 1). Phylogenetic analysis of adjacent genes revealed only a few instances of recent paralogs, which suggests that most of the duplications and subsequent positional gene arrangements are ancient. Three *P. macacae* subtelomeric regions have a nearly perfect synteny in *P. jirovecii* with the only difference being the absence of RG proteins in *P. jirovecii* (Supplementary Data 1). *P. oryctolagi* subtelomeres tend to be enriched in orphan genes that are not members of the Msg superfamily, and are of unknown function. *P. canis* subtelomeres are enriched in Msg-C family (see Msg section below). *P. wakefieldiae* subtelomeres are rich in *msg* genes, though their types are distinct from those of *P. carinii* and *P. murina*.

**Evolution of *msg* genes**. Up to 6% of the *Pneumocystis* genomes are comprised of copies of the *msg* superfamily, which are believed to be crucial mediators of pathogenesis through antigenic variation and interaction with the host cells. The superfamily is classified into five families A, B, C, D, and E based on protein domain architecture, phylogeny and expression mode[13,34,35]. The A family is the largest of the five, has been subdivided into three subfamilies (A1, A2, and A3) and is generally thought to contribute to antigenic variation. Their protein products contain cysteine-rich domain classified as N1 and M1 to M5.

To investigate the origin of *msg* genes, we used previously developed Hidden Markov Models[13] to search for corresponding gene models in the assemblies of *P. macacae*, *P. oryctolagi*, *P. canis*, and *P. wakefieldiae* and combined these data with published *msg* sequences annotated in *P. jirovecii*, *P. carinii*, and *P. murina* genomes[13,35]. Of note, in this study only a subset of *msg* genes were assembled for *P. oryctolagi*, *P. canis*, and *P. wakefieldiae* due to difficulties in assembling highly similar short reads from Illumina sequencing exclusively while a potentially complete set of *msg* genes were assembled for *P. macacae* using Illumina and Nanopore reads (Supplementary Table 3). The number of full-length *msg* genes available ranges from 9 in *P. oryctolagi* to 161 in *P. jirovecii*. Sequence-based clustering and phylogenetic analyses of all *msg* genes ($n = 482$) revealed that: (i) there is no evidence of interspecies transfer among *Pneumocystis* species (Fig. 4b to d; Supplementary Fig. 10), (ii) *msg* genes may have a polyphyletic origin, i.e., distinct families were present in most recent ancestors of *Pneumocystis* (Supplementary Fig. 10a), although convergent evolution of *msg* cannot be ruled out; (iii) *msg* genes experienced recombination early in their history as estimated by phylogenetic network analysis (Supplementary Fig. 10b and c).

While some gene expansions are relatively recent (for example, *msg* families A, C, and D) other expansions (*msg* families E and B) occurred before the emergence of *Pneumocystis* genus itself (Supplementary Fig. 11). Subsets of *msg* genes show strong host-specific sequence diversification (Fig. 4a), such as the current A family has emerged relatively recently at 43 mya (CI: 58–28 mya) compared to the emergence of the genus at 140 mya (see Methods; Supplementary Fig. 11). The A1 subfamily displays a substantial expansion in all species (Fig. 4a) and is subject to intraspecies recombination (Fig. 4b to d), which suggest that the most recent *Pneumocystis* ancestor may have developed a pre-Msg-A family,

which then evolved through duplication and recombination after the species separation.

The A3 subfamily has expanded only in clade P1 (especially in *P. jirovecii*) whereas A2 has expanded only in clade P2 (*P. carinii*, *P. murina* and to a lesser extent in *P. wakefieldiae*) (Fig. 4a). Although all members of the A family might have a shared deep ancestry, we found no evidence suggesting that the A1, A2, A3 subfamilies are directly derived from one another (Supplementary Fig. 10).

The *msg*-B family underwent a net independent expansion in *P. macacae* ($n = 10$) and *P. jirovecii* ($n = 12$), while being reduced to only one copy in *P. oryctolagi* and *P. canis*, and being completely absent in *P. wakefieldiae*, *P. carinii* and *P. murina* (Fig. 4a). Using Bayesian estimates, we estimated the origin of the B family to be older than the *Pneumocystis* genus itself (~211 vs. 140 mya; Supplementary Fig. 11).

The *msg*-D family is expanded only in *P. macacae* and *P. jirovecii*. The D family emerged at ~69 mya (CI: 109–40 mya) before the split of these two species (Supplementary Fig. 11), thus suggesting a role in adaptation to primates similar to the A3 subfamily. In contrast, the E family, which is conserved in all species, is much more ancient at ~311 mya (CI: 541–158 mya), again preceding the emergence of the genus (Supplementary Fig. 11).

*P. jirovecii* and *P. macacae* not only have a larger number of *msg*-associated cysteine-rich domains than other *Pneumocystis* species (Fig. 5a) but also a much greater sequence diversity per domain than other *Pneumocystis* species (Fig. 5c). Domain sequences cluster independently, with each cluster containing sequences from all *Pneumocystis* species (Fig. 5b). Domains M1 and M3 are more closely related to each other than other domains, which suggests a relatively recent duplication.

## Discussion

Surprisingly, analysis of core genomic regions of the nuclear genomes did not identify clear differences that are suggestive of mechanisms for host-specific adaptation; instead it is the highly polymorphic multicopy gene families (msgs) that appear to account for this adaptation. Msgs, which provide *Pneumocystis* with a mechanism for antigenic variation and consequent immune evasion, may have been important in allowing *Pneumocystis* organisms to infect mammals successfully, given that an adaptive immune system, which is critical to protection of mammals from exogenous pathogens, arose ~500 mya in the first jawed vertebrates[36]. Msgs likely played a dual role in avoiding the adaptive immunity and in cell adherence.

Based on our analysis, we propose the following series of events for the emergence and adaptation of *P. jirovecii* as a major human opportunistic pathogen (Fig. 6). First, there was a major shift of a pre-*Pneumocystis* lineage (possibly a soil- or plant-adapted organism) to mammals, which led to a genome reduction but with a proliferation of introns and expansions of cysteine-rich domain-containing proteins involved in immune escape and nutrient scavenging from hosts. *Pneumocystis* genomes encode multiple gene families that have experienced a rapid accumulation of mutations favoring fungal replication in mammals. Each *Pneumocystis* species has employed different strategies to adapt to their host including lineage-specific expansions of shared gene families such as *msg* A1, A3, and D in *P. jirovecii* or gain and expansion of RG proteins in *P. macacae*. In addition, some shared gene families also have acquired different properties (e.g., transmembrane domain and secreted signals) potentially contributing to host specificity. The absence of a reliable culture method and the inability to genetically manipulate *Pneumocystis* prevents

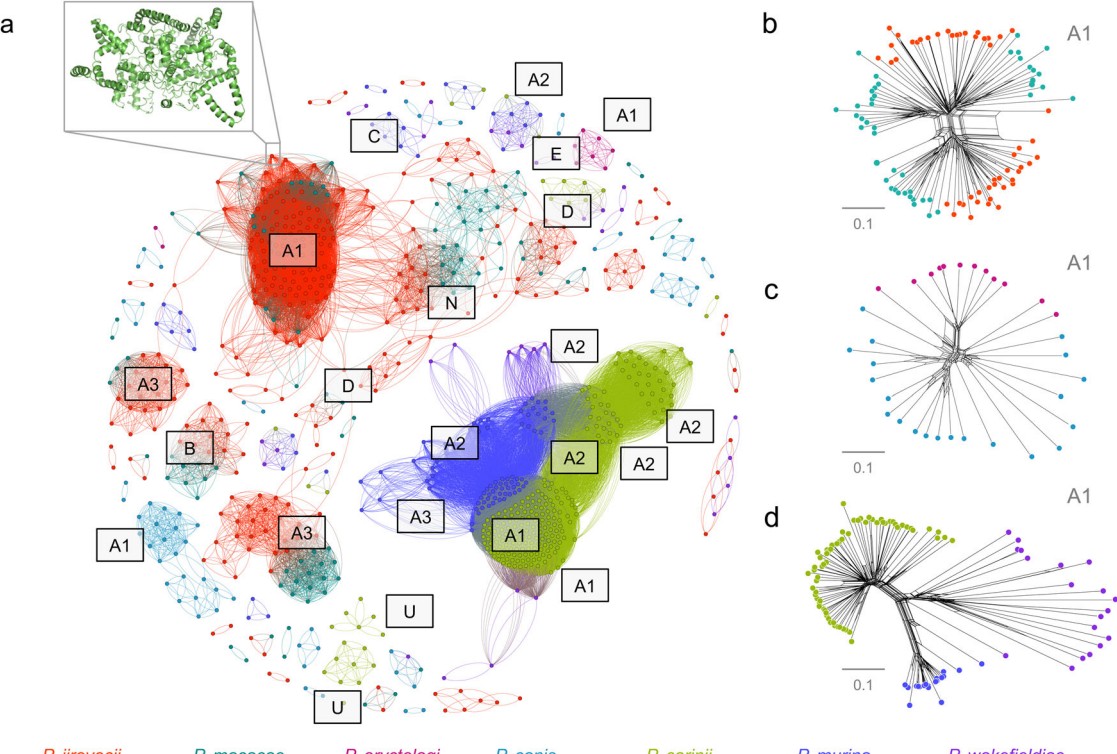

**Fig. 4 Clustering of *Pneumocystis* major surface glycoproteins (Msg). a** Graphical representation of similarity between 482 Msg proteins from seven *Pneumocystis* species generated using the Fruchterman Reingold algorithm. A 3-D model of a representative member of Msg-A1 protein family (NCBI locus tag T551_00910) generated using DESTINI is presented in the upper left insert. Individual protein sequences are shown as dots and color-coded by species as shown at the bottom of the figure. The edge between two dots indicates a global pairwise identity equal or greater than 45%. The letters represent Msg families (A to E) and subfamilies (A1 to A3). N and U letters represent potentially novel Msg sequences (relative to our prior study[35]) and unclassified sequences, respectively. For sake of clarity only the major clusters were annotated. **b** Phylogenetic network of a subset of Msg-A1 family ($n = 97$) in primate *Pneumocystis* including *P. jirovecii* (red) and *P. macacae* (dark cyan) suggesting recombination events at the root of the network. Nodes with more than two parents represent reticulate events. Bars represent the number of amino acid substitutions per site. **c** Phylogenetic network of Msg family A1 ($n = 33$) in *P. oryctolagi* (red violet) and *P. canis* (light blue). **d** Phylogenetic network of Msg-A1 family ($n = 113$) in rodent *Pneumocystis* including *P. carinii* (green), *P. murina* (dark blue), and P. wakefieldiae (blue violet). The network data are available at 10.5281/zenodo.4450766.

directly testing our model. Moreover, for the genes that we have now implicated in the process of host adaptation, only a few have been functionally characterized. Future studies on the role of these genes will be important to elucidate the molecular basis of host-specific adaptation by *Pneumocystis* pathogens.

## Methods

**Experimental design and *Pneumocystis* sample sources.** Animal and human subject experimentation guidelines of the National Institutes of Health (NIH) were followed in the conduct of this study. Studies of human and mouse *Pneumocystis* infection were approved by NIH Institutional Review Board (IRB) protocols 99-I-0084 and CCM 19-05, respectively. The collection and processing of a single human *P. jirovecii* sample from China (Pj55) was approved by the IRB of the First Affiliated Hospital of Chongqing Medical University, China (protocol no. 20172901). Written informed consent was obtained from the patient for participation in this study. The authors confirm that personal information was unidentifiable from this report. The National Institute of Allergy and Infectious Diseases (NIAID) Division of Intramural Research Animal Care and Use Program, as part of the NIH Intramural Research Program, approved all experimental procedures pertaining to the macaques (protocol LVD 26). Nonhuman primate study protocols were approved by the Institutional Animal Care and Use Committee of the University of California, Davis (protocol no. 7092), the Tulane National Primate Research Center (TNPRC), and the Institutional Animal Care and Use Committee (IACUC) (protocol no. P0351R). Studies of rabbit *Pneumocystis* infection were reviewed and approved by the Institutional Animal Care and Use Committee of the University of Michigan (protocol no. RO00008218). For rabbit samples obtained from France, the conditions for care of laboratory animals stipulated in European guidelines were followed (See: Council directives on the protection of animals for experimental and other scientific purposes, and J. Off.

Communautés Européennes, 86/609/EEC, 18 December 1986, L358). Samples from *Pneumocystis*-infected dog were collected as diagnostic samples and approved only for research purposes. The owner's consents for using samples and data were obtained on admission of the case and no further ethics permission was required because it was a routine diagnostic case and did not qualify as an animal experiment. Studies of rat *Pneumocystis* infection were approved by the Veteran Affairs animal protocol (VA ACORP #17-12-05-01). Clinical information and demographic data of the groups of individuals are presented in Supplementary Table 1. Three *P. jirovecii* samples were obtained as bronchoalveolar lavage from patients at the NIH Clinical Center in Bethesda, MD, USA and Chongqing Medical University in Chongqing, China. Six *P. macacae* samples were obtained as frozen lung tissues or formalin fixed paraffin embedded (FFPE) tissue sections prepared from SIV-infected rhesus macaques at the NIH Animal Center, Bethesda, Maryland ($n = 2$), the Tulane National Primate Research Center, Covington, Louisiana ($n = 3$), and the UC Davis California National Primate Research Center, Davis, California, USA ($n = 1$).

Four *P. oryctolagi* samples were obtained as frozen lung tissues from one rabbit with severe combined immunodeficiency at the University of Michigan, Ann Arbor, Michigan, USA, or as DNA from two corticosteroid treated rabbits and one rabbit with spontaneous *Pneumocystis* infection at the Institut Pasteur de Lille and the Institut National de la Recherche Agronomique de Tours Pathologie Aviaire et Parasitologie, Tours, France. *P. canis* samples were obtained as DNA from one Cavalier King Charles Spaniel dog at the University of Helsinki, Finland and one Whippet mixed-breed at the University of Veterinary Medicine, Vienna, Austria. The dogs were not laboratory animals. One *P. murina* sample was obtained from a heavily infected CD40L-KO mouse following a short-term in vitro culture. Genomic data obtained from *P. murina* isolates were combined with previously sequenced public data (Supplementary Table 2) and used for population genomics analysis (section "Speciation history of the *Pneumocystis* genus" and Supplementary Note 1). One frozen cell pellet and 4 agarose gel blocks containing *P. wakefieldiae* and *P. carinii* were obtained from immunosuppressed rats (one gel

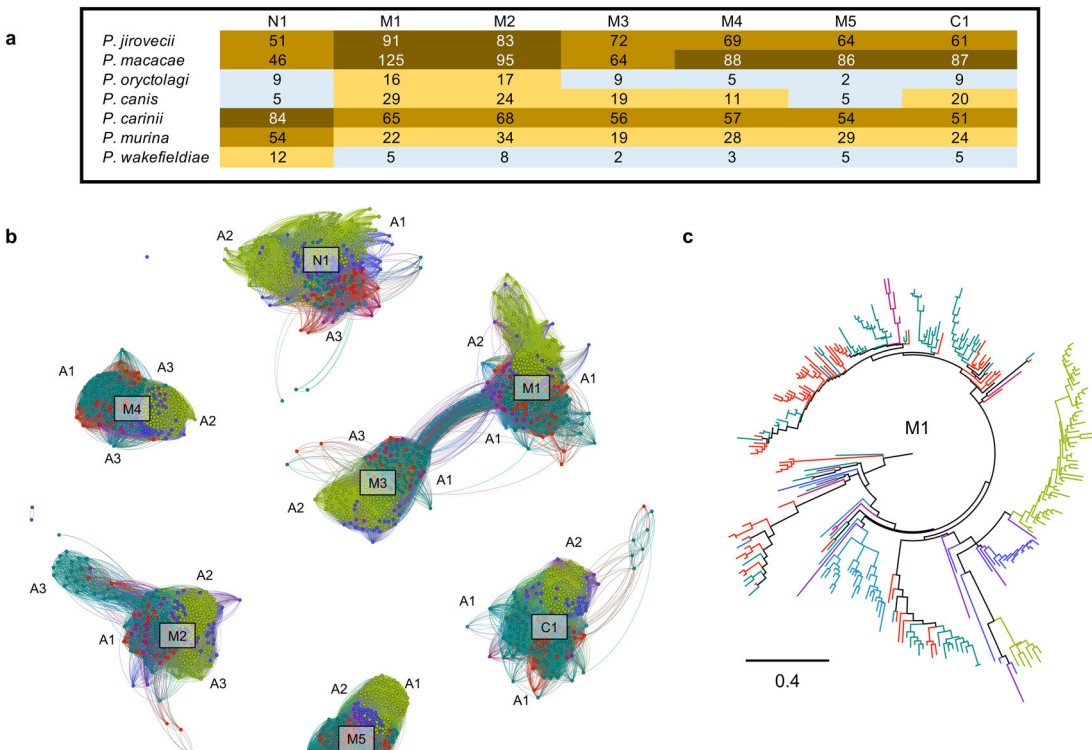

**Fig. 5 Evolution of Msg cysteine-rich protein domains in *Pneumocystis*. a** Heatmap showing the distribution of Msg domains in each *Pneumocystis* species. The color change from blue-orange-brown indicates an increase in the number of domains. **b** Graphical representation of protein similarity between domains, which highlights that the domains were present in the most recent common ancester and were maintained other than perhaps domains M1 and M3. Domains are clustered by a minimum BLASTp cutoff of 70% protein identity. **c** Maximum likelihood tree of the M1 domain. In both panels **b** and **c**, domains are color-coded by species as shown at the bottom.

block per rat) housed at the Cincinnati VA Medical Center, Veterinary Medicine Unit, Cincinnati, Ohio. Of note, all samples with low sequence coverage were used either in combination with other samples to generate consensus genome assemblies or for population genetic analyses in which variation in sequencing coverage are explicitly accounted for.

**Genome sequencing, assembly, and annotation**. Genomic DNA in agarose gel blocks was extracted using the Zymoclean Gel DNA Recovery Kit (Zymo Research). Genomic DNA in FFPE sections was extracted using the AllPrep DNA/ RNA FFPE Kit (Qiagen). Genomic DNA in frozen lung tissues from two *P. macacae*-infected macaques and a single *P. oryctolagi*-infected rabbit was treated with a sequential collagenase type I and DNase I digestion[13] to deplete host DNA and extracted using the MasterPure Yeast DNA purification kit (Epicentre Bio-technologies, Madison, WI, USA). Genomic DNA in bronchoalveolar lavage samples from *P. jirovecii*-infected patients was extracted using the MasterPure Yeast DNA purification kit. Total RNAs for *P. macacae*, *P. wakefieldiae* and *P. murina* were isolated using RNeasy Mini kit (Qiagen). For DNA samples with small quantity, including three *P. oryctolagi* samples (RABF, RAB1, and RAB2B) and one *P. jirovecii* sample (RU817), we performed whole-genome amplification prior to Illumina sequencing. Five microliters of each DNA sample were amplified in a 50-ul reaction using an Illustra GenomiPhi DNA V3 DNA amplification kit (GE Healthcare, United Kingdom). Genomic DNA samples were quantified using Qubit dsDNA HS assay kit (Invitrogen) and NanoDrop (ThermoFisher). RNA integrity and quality were assessed using Bioanalyzer RNA 6000 picoassay (Agi-lent). The identities of *Pneumocystis* organisms were verified by PCR and Sanger sequencing of mtLSU before high throughput sequencing. No quantitative PCR methods were used. For most of the DNA samples, at least one microgram of each DNA or RNA (depleted of ribosomal RNA using Illumina Ribo-Zero rRNA Removal Kit) sample was sequenced commercially using the Illumina HiSeq2500 platform with 150 or 250-base paired-end libraries (Novogene Inc, USA) or for one DNA sample of *P. jirovecii* from a Chinese patient using a single-read SE50 library using the MGISeq 2000 platform (MGI Tec, China).

Adapters and low-quality reads were discarded using trimmomatic v0.36[37] with the parameters "-phred33 LEADING:3 TRAILING:3 SLIDINGWINDOW:4:15 MINLEN:36". Host DNA and other contaminating sequences were removed by

mapping against host genomes using Bowtie2 v2.4.1[38]. Filtered Illumina reads were assembled de novo using Spades v3.11.1[39]. Details for host DNA sequences removal, *Pneumocystis* reads recovery and de novo assembly protocols are presented in Supplementary Methods. Completeness of assemblies was estimated using BUSCO v9[40], FGMP v1.0.1[41] and CEGMA v2.5[42]. For *P. macacae*, in addition to Illumina sequencing, Nanopore sequencing was performed on *P. macacae* DNA samples prepared from a single heavily infected macaque (P2C) with ~68% *Pneumocystis* DNA based on prior Illumina sequencing (Supplementary Table 2). High molecular weight genomic DNA fragments were isolated using the BluePippin (Sage Science) with the high-pass filtering protocol. A DNA library was prepared using the rapid Sequencing kit (SQK-RAD0004) from Oxford Nanopore Technologies (Oxford, UK) and loaded in the MinION sequencing device. Host reads were removed by mapping to the Rhesus macaque genome (NCBI accession number GCF_000772875.2_Mmul_8.0.1) using Minimap2 v2.10[43]. Unmapped reads were aligned to the draft version of *P. macacae* assembly built previously using Illumina data (Supplementary Methods) with ngmlr v0.2.7[43]. A total of 1,633,376 Nanopore reads were obtained, of which ~5% were attributed to *Pneumocystis* (27-fold coverage), which is much less than the 68% based on Illumina data (Supplementary Table 2). This suggests that many *P. macacae* genomic DNA fragments were too short to pass the size selection filter. *Pneumocystis* Nanopore reads were assembled using Canu v.1.8.0[44], overlapped with the assembly using Racon v.1.3.3[45] and polished with Pilon v1.22[46] using the Illumina reads aligned with BWA MEM v0.7.17[47].

Illumina RNA-Seq of the *P. macacae* sample P2C yielded 22 million reads, of which ~92% were attributed to *Pneumocystis* (Supplementary Table 2). Filtered reads were mapped to the *P. macacae* assembly using hisat2 v2.2.0[48], sorted with SAMtools v1.10[49] and filtered with PICARD v2.1.1 (http://broadinstitute.github.io/ picard). De novo transcriptome assembly of filtered reads was performed with Trinity[50]. Quantification of transcript abundance was performed using Kallisto v0.46.1[51]. *P. wakefieldiae* (2A) and *P. murina* RNA-Seq data were processed similarly (Supplementary Table 2). DNA transposons, retrotransposons and low complexity repeats were identified using RepeatMasker[52], RepBase[53] and TransposonPSI (http://transposonpsi.sourceforge.net). *Pneumocystis* telomere motif "TTAGGG"[16] was searched using "FindTelomere" (available at https:// github.com/JanaSperschneider/FindTelomeres). The genomes of *P. carinii* strain Ccin[14] and strain SE6[12] were scaffolded with Satsuma[54] using the *P. carinii* strain

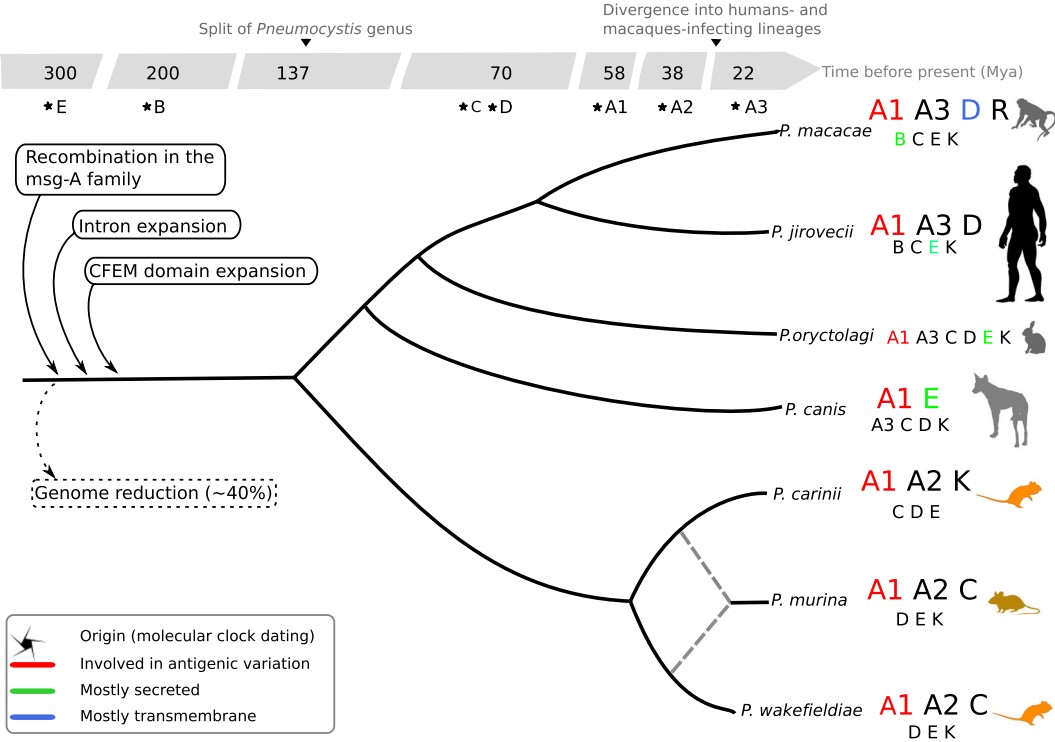

**Fig. 6 Overview of the genomic evolution of the *Pneumocystis* genus.** Gene families are represented by letters: A to E for the five families of major surface glycoproteins (Msg) with the A family being further subdivided into three subfamilies A1, A2, and A3; K and R for kexins and arginine-glycine rich proteins, respectively. Larger fonts indicate expansions as inferred by maximum likelihood phylogenetic trees and networks. Dashed lines represent ancient hybridization between *P. carinii* and *P. wakefieldiae*. Detailed analysis also reveals distinct phylogenetic clusters within subfamilies. Introns and CFEM (common in fungal extracellular membrane) domains are enriched in *Pneumocystis* genes which indicate that these elements were likely present in the most recent common ancestor of *Pneumocystis* species. Animal icons were obtained from http://phylopic.org under creative commons licenses https://creativecommons.org/licenses/by/3.0/: mouse (Anthony Caravaggi; license CC BY-NC-SA 3.0); dog (Sam Fraser-Smith and vectorized by T. Michael Keesey; license CC BY 3.0), rabbit (by Anthony Caravaggi; license CC BY-NC-SA 3.0), and rat (by Rebecca Groom; license CC BY-NC-SA 3.0). Icons original black color background were modified to gray and orange colors.

B80 as reference genome[13]. *P. macacae*, *P. oryctolagi*, *P. canis* Ck1, *P. canis* A, *P. wakefieldiae*, and *P. carinii* (strains Ccin and SE6) genome assemblies were annotated using Funannotate v1.5.3 (https://doi.org/10.5281/zenodo.1134477). The homology evidence consists of fungal proteins from UniProt[55] and BUSCO v9 fungal proteins[40]. For *P. macacae* and *P. wakefieldiae*, RNA-Seq mapping files (BAM) and de novo transcriptome assemblies were used as hints for AUGUSTUS[56]. Ab initio predictions were performed using GeneMark-ES[57]. All evidences were merged using EvidenceModeler[58]. *Taphrina* genomes (*T. deformans*, *T. wiesneri*, *T. flavoruba*, and *T. populina*[32,59]) and *P. canis* Ck2 genome were annotated using MAKER2[60] because predicted gene models showed a better quality than those obtained from Funannotate. MAKER2 integrates ab initio prediction from SNAP[61], AUGUSTUS built-in *Pneumocystis* gene models[62] and GeneMark-ES as well as BLAST- based homology evidences from a custom fungal proteins database. GPI prediction was performed using PredGPI[63], big-PI[64] and KohGPI[65]. Signal peptide leader sequences and transmembrane helices were predicted using Signal-P version 5[66] and TMpred[67], respectively. Protein domains were inferred using Pfam database version 3.1[68] with PfamScan (https://bio.tools/pfamscan_api). PRIAM[69] release JAN2018 was used to predict ECs using the following options: minimum probability > 0.5, profile coverage > 70%, check catalytic—TRUE and e-value < 10$^{-3}$. *Pneumocystis* mitochondrial genome assembly and annotations are presented in Supplementary Methods. Three dimensional (3D) protein structure prediction of Msg proteins was performed using DESTINI[70] and visualized with PyMol (www.pymol.org).

**Comparative genomics.** All genomes were pairwise aligned to the *P. jirovecii* strain RU7 genome NCBI accession GCF_001477535.1[13] using LAST version 921[22] with the MAM4 seeding scheme[71]. One-to-one pairwise alignments were created using maf-swap utility of LAST package and merged into a single multispecies whole-genome alignment using LAST's maf-join utility. Pairwise rearrangement distances in terms of minimum number of rearrangements were inferred using GRIMM[72] and Mauve[73]. Breakpoints of genomic rearrangements were refined with Cassis[74] and annotated using BEDtools[75] 'annotate' command. Average pairwise genome-wide nucleotide divergences were computed with Minimap2[76]. Synteny

visualization was carried out using Synima[77]. Msg protein similarity networks were based on global pairwise identity obtained from pairwise alignments of full-length proteins using Needle[78] or BLASTp[79] identity scores for individual protein domains. The networks presented in Figs. 4 and 5 were generated using the Fruchterman Reingold algorithm as implemented in Gephi v0.9.2[80]. To generate genome sequences for low coverage samples, raw Illumina reads were aligned to reference genomes with LAST. Resulting alignments were filtered and sorted with SAMtools. SNPs and indels were identified, normalized, filtered and used to generate consensus genomes using bcftools[81]. Pairwise divergence scores were computed using minimap2. Sequence motifs were visualized using WebLogo[82]. Multi-panel figures were assembled in Inkscape (https://inkscape.org).

To investigate the evolution of introns in *Pneumocystis* species, we identified unambiguous one-to-one orthologous clusters using reciprocal best Blast hit (e-value of 10$^{-10}$ as cutoff) in seven *Pneumocystis* species as well as in three other Taphrinomycotina fungi: *S. pombe*, *T. deformans* and *N. irregularis*. Intron position coordinates were extracted from annotated genomes using Replicer[83] and projected onto protein multiple alignments using custom scripts. Homologous splice sites in annotated protein sequence alignments were identified using MALIN[84]. We required at least 11 unambiguous splicing sites and five minimal nongapped positions. A potential splice was considered unambiguous if the site has at least five nongap positions in the aligned sequences in both the left and right sides. MALIN uses a rates-across sites markov model with branch specific gain and loss rates to infer evolution of introns. Gain and loss rates were optimized through numerical optimizations. Fungi have a strong tendency to intron loss with few exceptions (e.g., *Cryptococcus*) whereas gain of intron is relatively rare. Thus, we penalized intron gain and set the variation rate to 4/3 for loss and gain levels. Intron evolutionary history was inferred using a posterior probabilistic estimation with 100 bootstrap support values.

**Phylogenomics.** Orthologous gene families were inferred using OrthoFinder v.2.3.11[85]. In addition to *Pneumocystis* and *Taphrina* species, the predicted proteins for the following species were downloaded from NCBI: *Neolecta irregularis* (accession no. GCA_001929475.1), *S. pombe* (GCF_000002945.1), *S. cryophilus* (GCF_000004155.1),

*S. octosporus* (GCF_000150505.1), *S. japonicus* (GCF_000149845.2), *Saitoella complicata* (GCF_001661265.1), *Neurospora crassa* (GCF_000182925.2), *Cryptococcus neoformans* (GCF_000149245.1), *Rhizopus oryzae* (GCA_000697725.1) and *Batrachochytrium dendrobatidis* (GCF_000203795.1). Single-copy genes were extracted from OrthoFinder output ($n = 106$) and concatenated into a protein alignment containing 458,948 distinct alignment patterns (i.e., unique columns in the alignment) with a gap proportion of 12.2%. Maximum likelihood tree analysis was performed using RAxML v 8.2.5[86] with 1000 bootstraps as support values. The LG model[87] was selected as the best amino-acid model based on the likelihood PROTGAMMAAUTO in RAxML. One hundred and six gene trees were estimated from each of the single-copy genes. The Shimodaira–Hasegawa test[18] was performed on the tree topology for each of the gene trees and the concatenated alignment using IQ-Tree[88] with 1000 RELL bootstrap replicates. IQ-Tree was run as follows: "iqtree -s {input.phy} -z {input.t} -n 0 -zb {params.n}", where {input.phy} represents the concatenated alignment of 106 genes in phylip format (24 species; 543,202 amino-acid sites with 14.2076% of constant sites), {input.t} represents a file containing individual maximum likelihood phylogenetic trees for each of the 106 genes, -n 0 parameter avoid tree search and estimate model parameters based on an initial parsimony tree (the best-fit model according to BIC was LG + F + R7) and the -zb option specifies the number of bootstrap replicates for the resampling estimated log-likelihood method (RELL).

To infer the species phylogeny using mitochondrial genomes, protein-coding genes were extracted, aligned using Clustal Omega[89], and concatenated. The resulting alignment was used to infer phylogeny using IQ-Tree v1.6[90] with TVM + F + I + G4 as the Best-fit substitution model and 1000 ultrafast bootstraps and SH-aLRT test. A total of 33 mitogenomes from seven *Pneumocystis* species were used: *P. jirovecii* [$n = 18$ including 3 sequences from this study and 4 from previous studies[5,12,17], *P. macacae* ($n = 4$), *P. oryctolagi* ($n = 4$), *P. canis* [$n = 4$, refs. 17,91], *P. carinii* [$n = 2$, refs. 17,91], *P. murina* [$n = 1$, ref. 17], and *P. wakefieldiae* ($n = 1$). Phylogenetic reconciliations of species tree and gene trees were performed using Notung[92]. Ancestral reconstruction of gene family's history was performed using Count[93]. Phylogenetic network for Msg protein families was inferred using SplitTree[94]. The detection of putative mosaic genes was performed using TOPALi v2.5[95].

**Phylodating**. Single-copy orthogroup nucleotide sequences were aligned using MACS v0.9b1[96]. Highly polymorphic *msg* sequences were excluded using BLASTn[79] with an e-value of $10^{-5}$ as cutoff against 479 published *msg* sequences[13]. We inferred the divergence timing using two datasets: (1) 24 single-copy nuclear gene orthologs shared by all *Pneumocystis* and *S. pombe*; and (2) 568 nuclear genes found in all *Pneumocystis* species. BEAST inputs were prepared using BEAUTi v2.5.1[97]. Unlinked relaxed lognormal molecular clock models[98,99] and calibrated birth-death tree priors[100] were used to estimate the divergence times and the credibility intervals. The substitution site model HKY was applied[101]. Three secondary calibration priors were used: (i) *P. jirovecii*/*P. macacae* divergence with a median time of 65 mya as 95% highest posterior density (HPD)[5], (ii) the emergence of the *Pneumocystis* genus at a minimum age of 100 mya[4], and (iii) the *Schizosaccharomyces*—*Pneumocystis* split at ~467 mya[102]. We sampled from various priors and found a minor difference between the marginal posterior distribution on rate and the marginal prior distribution. This indicates that the posterior simply reflects the prior. For the dataset 2, the 568 gene alignments were concatenated in a super alignment with 568 partitions, with each partition defined by one gene. Gene partitions were collapsed using PartitionFinder v2.1.1[103] with the "greedy" search to find optimal partitioning scheme. The alignment was split in three partitions in BEAST. Three independent runs for each dataset were performed separately for 60 million generations using random seeds. Run convergence was assessed with Tracer v1.7.1 (minimum effective sampling size of 200 with a burn-in of 10%). Trees were summarized using TreeAnnotator v.2.5.1 (http://beast.bio.ed.ac.uk/treeannotator) and visualized using FigTree v.1.4.4 (http://tree.bio.ed.ac.uk/software/figtree) to obtain the means and 95% HPD. Host divergences were obtained from the most recent mammal tree of life[6], available at http://vertlife.org/data/mammals. The dating of fungal gene families was performed similarly. Phylogenetic trees with geological time scale were visualized using strap version 1.4[104].

**Population genomics**. Sequence data sources and primary statistics are presented in Supplementary Table 2. Adapter sequences and low-quality headers of base sequences were removed using Trimmomatic[37]. Interspecies read alignment was performed using LAST[22] with the MAM4 seeding scheme[71]. Alignments were processed by last-split utility to allow interspecies rearrangements, sorted using SAMtools v1.10[49]. Duplicates were removed using PICARD v2.1.1. To compute the $F_{ST}$ and nucleotide diversity (Watterson, pairwise, FuLi, fayH, L), we calculated the unfolded site frequency spectra for each population using the Analysis of Next Generation Sequencing Data (ANGSD)[23]. Site frequency spectra was estimated using ANGSD[23,105]. Hierarchical clustering was performed using ngsCovar[106]. All data were formatted to fit a sliding windows of 1-10 kb using BEDTools[107]. For each window, an average value of the statistics was calculated using custom scripts.

**Gene flow inference**. To infer a phylogenetic network, we used 1718 one-to-one orthologs from gene catalogs of seven *Pneumocystis* species using reciprocal best BLASTp hit with an e-value of $10^{-10}$ as cutoff. Sequences from each orthologous group were aligned using Muscle[108]. Alignments with evidence of intragenic recombination were filtered out using PhiPack[109] with a *p*-value of 0.05 as cutoff. For each aligned group a maximum likelihood (ML) tree was inferred using RAxML-ng[110] with GTR + G model and 100 bootstrap replicates, and Bayesian tree was generated using BEAST2[97]. ML trees were filtered using the following criteria: 0.9 as the maximum proportion of missing data, 100 as the minimum number of parsimony-informative sites, 50 as the minimum bootstrap node-support value and 0.05 as the minimum *p*-value for rejecting the null hypothesis of no recombination within the alignment. BEAST trees with an effective sampling size <200 were removed. Filtered trees were summarized using Treannotator (https://www.beast2.org/treeannotator/). Summary trees with an average posterior support inferior to 0.8 were discarded. Species network was inferred using PhyloNet option "InferNetwork_MPL"[24] with prior reticulation events ranging from 1 to 4. Phylogenetic networks were visualized using Dendroscope 3[111].

The highest probability network inferred a hybridization between *P. carinii* and *P. wakefieldiae* leading to *P. murina* followed by a backcrossing between *P. murina* with *P. wakefieldiae* (log probability = −12759.4). Analysis of tree topology frequencies revealed that 64% of the trees were consistent with the topology of (*P. carinii*, (*P. murina*, *P. wakefieldiae*)), 28% with the topology of (*P. wakefieldiae*, (*P. carinii*, *P. murina*)) and 8% with the topology of (*P. murina*, (*P. carinii*, *P. wakefieldiae*)).

**Detection of positive selection**. To search for genes that have been subjected to positive selection in *P. jirovecii* alone after the divergence from *P. macacae*, we used the branch-site test[33] as implemented in PAML[112], which detects sites that have undergone positive selection in a specific branch of the phylogenetic tree (foreground branch). A set of 2466 orthologous groups between *P. jirovecii*, *P. macacae* and *P. oryctolagi* was used for the test. $d_N/d_S$ ratio estimates per branch per gene were obtained using Codeml (PAML v4.4c) with a free ratio model of evolution. This process identified 244 genes with a signal of positive selection only in *P. jirovecii* ($d_N/d_S > 1$).

**Statistics and reproducibility**. All analyses were conducted in R version 3.3.2[113]. Statistical details and experimental design for different data analyses are presented in the respective results and methods sections. No sample-size calculation was performed for genome sequencing. The assessment of protein domains associated Gene Ontologies (GO) term enrichment was performed using hypergeometric test as implemented in dcGOR version 1.0.6[114] (*p*-values adjusted by Benjamini–Hochberg method). The statistical significance of differences among groups was determined using Wilcoxon signed-rank test.

**Reporting summary**. Further information on research design is available in the Nature Research Reporting Summary linked to this article.

## Data availability

The datasets generated during and/or analyzed during the current study are available at NCBI BioProject portal (https://www.ncbi.nlm.nih.gov/bioproject/): *Pneumocystis macacae* strain P2C (no. PRJNA632025); *P. oryctolagi* strain CS1 (PRJNA632560); *P. canis* strain Ck1 (PRJNA632556); *P. canis* strain Ck2 (PRJNA632878); *P. canis* strain A (PRJNA636786);*P. wakefieldiae* strain 2A (PRJNA632570); *P. jirovecii* strain 55 (PRJNA647920), *P. jirovecii* strain 54c (PRJNA648092), *P. jirovecii* strain 46 (PRJNA648096), *P. macacae* strain CJ36 (PRJNA648103), *P. macacae* strain ER17 (PRJNA648108), *P. macacae* strain UC86 (PRJNA648112), and *P. macacae* strain GL92 (PRJNA648115).

## Code availability

All custom bioinformatic analyses were conducted using Perl v5.26.0 (http://www.perl.org/) or Python v.3.6 (http://www.python.org) scripts. Pipelines were written with Snakemake v5.11.2[115]. Custom codes generated for this project are available at GitHub: https://github.com/ocisse/pneumocystis_evolution and a stable released version is available at https://doi.org/10.5281/zenodo.4450766.

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

## Acknowledgements

This work has been funded in whole or in part with federal funds from the Intramural Research Program of the US National Institutes of Health (NIH) Clinical Center and the National Institute of Allergy and Infectious Diseases (NIAID). This study used the Office of Cyber Infrastructure and Computational Biology (OCICB) High Performance Computing (HPC) cluster at the National Institute of Allergy and Infectious Diseases (NIAID), Bethesda, MD. This study also utilized the high-performance computational capabilities of the Biowulf Linux cluster at the NIH, Bethesda, MD (http://biowulf.nih.gov). The content of this publication neither necessarily reflect the views or policies of the Department of Health and Human Services, nor does mention of trade names, commercial products, or organizations imply endorsement by the U.S. Government. M.T.C. is a VA Senior Research Career Scientist supported by 5IK6BX005232. Her lab is funded by support from the NIH R01HL146266 and VA Grant I01BX004441. J.E.S and C.A.C. are CIFAR Fellows in the program Fungal Kingdom: Threats and Opportunities. Animal icons used in Figs. 3 and 6 were obtained from http://phylopic.org under creative commons licenses https://creativecommons.org/licenses/by/3.0/.

## Author contributions

O.H.C., L.M. and J.A.K. conceived the project and designed all the experiments. L.M., O.H.C., C.W.L., J.B., J.X., J.S., R.B., B.P., K.V.R., R.K., A.S., M.C., V.H., J.C., L.P., M.T.C., G.K., Y.L. and J.A.K. performed the laboratory work to obtain samples for sequencing. O.H.C., L.M., J.P.D., P.P.K. and J.L. developed and implemented methods for sample processing, library preparation and sequencing. O.H.C., L.M., J.E.S., C.A.C. and N.S.U. analyzed the data. O.H.C., L.M. and J.A.K. drafted the manuscript, which was revised by all authors.

## Funding

## Competing interests

The authors declare no competing interests.
