## [Peer Review File · Communications Biology]

Reviewers' comments:

Reviewer #1 (Remarks to the Author):

Review:

The manuscript by Cissé et al. entitled "Genomic insights into the host specific adaptation of the *Pneumocystis* genus and emergence of the human pathogen *Pneumocystis jirovecii*" is very well written and clear. All data are accessible for the researchers, and methods as well as statistics are well described and plausible.

Comprehensive research regarding *Pneumocystis* species is still sparse. This article allows a deep insight in the genetics but also possible biochemical processes in five novel *Pneumocystis* species (*P. jirovecii*, *P. macacae*, *P. oryctolagi*, *P. carinii*, and *P. wakefieldiae*) from different mammalian hosts. Within this article, the complete genomes of the four animal-derived species were analysed for the first time and were compared to the three previously published genomes of *P. jirovecii* strains, *P. carinii*, and *P. murina*. The methods used are comprehensive and included analysis of the genomic, as well as mitochondrial genes, transcriptome analysis, and phylogenetic dating. This outstanding manuscript allows a closer insight in the complex genetic rearrangements done by *Pneumocystis* for adaptation to the host species, which is implicitly necessary for understanding *Pneumocystis* spp. and *Pneumocystis* pneumonia.

I only have minor comments which should be best addressed with additional discussion points rather than additional work.

Content-related comments:

The authors used four different DNA and/or DNA/RNA extraction methods and also three different next generation sequencing machines (Illumina HiSeq2500, MinION, and MGISEq 2000 platform) for the samples. It is not easy to collect *Pneumocystis*-positive samples from animals. Therefore, I suspect, it was due to the worldwide collection of samples at various time points. Additionally, the samples used were quite different in quality (BALF, fresh lung tissue, paraffin-embedded lung tissue). Therefore, it is necessary to adjust sample size and DNA extraction protocols. Please, mention the sample sizes (g lung sample or mL BALF) used for DNA extraction in the text or supplementary table 1 or 2. Possibly, one could argue that the variable extraction methods in combination with a variance of preparation protocols and three high throughput sequencing machines with different sequencing principles and precisions might lead to a bias in the data. Please add a short explanation.

How would you explain those samples with low numbers of *Pneumocystis* reads? If we expect that the patients/animals had an active PCP with high numbers of *Pneumocystis* per gram of sample, the samples should have higher numbers of *Pneumocystis* reads. I would just add one or two additional sentences for that.

All animals, despite three rabbits, were laboratory animals, where the (clinically proven?) PCP was induced by immunosuppression of the animals. Following questions to answer to this topic:

A) Please, state whether the dogs were laboratory animals or not.

B) The authors mentioned that they tested all samples with mtLSU rRNA PCR and sanger sequencing prior high throughput sequencing. Was it just for confirmation of *Pneumocystis* in the samples? Or did the authors also quantify the *Pneumocystis* organism numbers (equivalents) in the samples with qPCR methods? Even if the copy numbers of e.g. mtLSU gene(s) are different in the animal-derived *Pneumocystis*, it would be helpful to assess the amount of *Pneumocystis* DNA in the samples. Due to the normalization of samples during processing for high throughput sequencing, one cannot quantify the amount of *Pneumocystis* in a DNA extract from a specific sample size. But, I guess, PCR quantification of *Pneumocystis* present in the samples, might help to interpret the relatively low amounts of *Pneumocystis* reads in some samples (see Suppl. Table 2, e.g. *P. jirovecii* Pj46 and Pj54c or *P. macacae* CJ36, etc.). Possibly, the patients/animals did not suffer from an active PCP, but were colonized?

C) Possibly, one of the DNA extraction kits had a lower extraction efficiency either for *Pneumocystis* or for whole DNA than the other kits used? This might also explain some of the samples with low *Pneumocystis* reads. Please explain.

Additional comments regarding text and graphics:

- Page 23, line 509: change "...obtained France..." to "...obtained from France..."

- Page 23, line 513: change "...approved for only for research..." to "...approved only for research..."
- Page 37, line 813: write "humans" instead of "Humans"
- Page 42, Fig. 3: the font on the right side is too small to read. Possibly it is better to switch the complete Figure to landscape
- Supplementary material, page 55, line 1102: the font size changes in the last word "using"
- Supplementary material, page 56, line 1104, 1106, and 1110: the font size changes in the words/phrases "Retrieved reads...", and two times the word "using"
- Supplementary material, page 62, line 1258: there is a doubled period
- Supplementary material, page 64, line 1286: correct "metablism" to "metabolism"
- Supplementary material, page 66, Figure S1: the font size in the figure is way too small for reading, possibly switch the graphs to landscape and enlarge the font
- Supplementary material, page 69, legend of Figure S3: "Fixation" should be written uncapitalized
- Supplementary material, page 74, Figure S5: Figures a, b -, and possibly also d are too small for reading the legends / the text, which might be important for understanding the graphs

With best regards,

Reviewer #2 (Remarks to the Author):

In this manuscript the authors sequenced the genome of *Pneumocystis* infecting macaques, rabbits, dogs and rats and compared these genomes to those of species infecting humans, mice and rats. The authors inferred a phylogeny from orthologous genes, identified important genetic features for host adaptation and estimated speciation times relative to the rise of their mammalian hosts.

This article is technically sound and contributes to our understanding of *Pneumocystis* adaptation and evolution. Clearly, it is a relevant research manuscript.

As mentioned by the authors, there are two unexpected finding regarding the phylogeny shown in Figure 2. One of them is the association between *P. wakefieldiae* and *P. murina* and the other is the association between *P. oryctolagi* and the common ancestor of *P. jirovecii* and *P. macacae*.

The evidence supporting the proximity between *P. wakefieldiae* and *P. murina* is quite convincing (the bootstraps, the posterior probabilities, the percentage of trees supporting this association, the genome structure and the mitochondrial phylogeny). And the explanation given to the placement of *P. oryctolagi* (host switching) is perhaps the most parsimonious one. If the authors would like to strengthen further their conclusions they could make a parametric bootstrap analysis to rule out that the unexpected phylogeny is due to long branch attraction (LBA). However, I don't think this analysis is mandatory by any means (i.e. by looking at the tree from figure 2 there seems to be no LBA). Anyway, there is a nice explanation of the parametric bootstrap in: Baum & Smith "Tree Thinking: an introduction to phylogenetic biology" (2013) if the authors think this analysis could complement their results.

Regarding the phylodating, it is not clear for me why the authors used as a calibration prior a median time of 65 mya for the divergence between *P.jirovecii*/*P.macacae* since the divergence between human and macaque is estimated to be approximately 20 mya. Further explanation in the manuscript would make the use of this prior more clear for the readers. In addition, it is important that the authors evaluate the impact of the priors in their results by running the analyses without the sequences (i.e. sampling from priors) and to test whether sequences are informative on estimated divergence dates (see Drummond & Bouckaert "Bayesian Evolutionary Analysis with Beast" (2015)).

Minor comments

Give more details on how the Shimodaira-Hasewaga test was used. The readers would appreciate this.

In line 394 the authors write that "msg genes may have a polyphyletic origin". Although the authors clarify this by later writhing that "i.e., distinct families were present in the most recent ancestors of Pneumocystis" this may be a bit confusing. Do the authors mean that msg really originated independently or that they originated once and diversified before the last common ancestor of extant Pneumocystis species?

Lines 481 and 482. The authors used the word "further" three times in a single sentence.

Luis Delaye

Reviewer #3 (Remarks to the Author):

This study is impressive and provides new data on the evolution of Pneumocystis and its adaptation to hosts.

- L63 : this is not fully true in rodents since different rodent species can be infected by the same Pneumocystis species. PMID: 29117878 ; PMID: 31247340. Corresponds to ref 10, 11. Please rephrase
- Line 148-150: no fully true regarding ref 10 and 11. Other hypothesis?
- Can high variability of *P. macaque* among individuals be explained by the use of Nanopore which is known to introduce technical errors. Illumina reads should be able to correct some of these errors. More details about the animals should be interesting to provide to understand the diversity obtained. Only consensus sequence has been used? Only one of the 6 sequences?
- It would be important (does not exist in the literature) to compare interindividual variation as compared to interspecies variation. Indeed, the authors found a high (12%) variation % in 6 macaques which can be corresponding to different species as well. Please add an analysis regarding this to understand how individual Pneumocystis varies inside the species as compared to the inter species variation.

Response to Reviewers' Comments

Genomic insights into the host specific adaptation of the *Pneumocystis* genus and emergence of the human pathogen *Pneumocystis jirovecii* by Cisse *et al.*

Manuscript # COMMSBIO-20-2703-T

General comments:

- We've made minor edits to the text to improve the clarity of the document (track change option activated).
- The name "*P. carinii* forma specialis *macacae*" has been replaced by "*Pneumocystis sp. macacae*", which is now the validated taxonomic name (NCBI taxon id 2698480). Similarly, the name "*P. carinii* f. sp. *canis*" has been replaced by the now validated "*P. canis*" (NCBI taxon id 2698477).
- When noted the lines or pages of the revision refer to the manuscript with tracked changes.

Reviewers' comments:

Reviewer #1 (Remarks to the Author):

Content-related comments:

The authors used four different DNA and/or DNA/RNA extraction methods and also three different next generation sequencing machines (Illumina HiSeq2500, MinION, and MGISEq 2000 platform) for the samples. It is not easy to collect *Pneumocystis*-positive samples from animals. Therefore, I suspect, it was due to the worldwide collection of samples at various time points. Additionally, the samples used were quite different in quality (BALF, fresh lung tissue, paraffin-embedded lung tissue). Therefore, it is necessary to adjust sample size and DNA extraction protocols. Please, mention the sample sizes (g lung sample or mL BALF) used for DNA extraction in the text or supplementary table 1 or 2. Possibly, one could argue that the variable extraction methods in combination with a variance of preparation protocols and three high throughput sequencing machines with different sequencing principles and precisions might lead to a bias in the data. Please add a short explanation.

Response: We appreciate these thoughtful comments. However, we are unable to provide the requested information. For FFPE samples we do not have weights of the slices utilized (and weights would not accurately reflect the embedded tissue), and for BAL samples, we utilized residual samples available after processing by our clinical labs. We did not weigh the lung tissue used. In addition, DNA rather than tissue/BAL was provided by collaborators for some samples.

We do not feel that standardizing based on weight/ml would be meaningful. Since we are unable to culture any *Pneumocystis* species, and thus obtain relatively pure organisms' populations, the most important determinant in recovery of *Pneumocystis* DNA from the various samples is the intensity of infection, which is highly variable, from relatively light infection in immunocompetent hosts (who may be colonized/have

subclinical infection), to immunodeficient hosts with typically heavier levels of infection. Thus, the amount of input tissue per se is not helpful in standardizing the amount of *Pneumocystis* DNA being processed.

We do not think that different levels of sampling led to sequencing bias. We used optimal methods for extracting DNA from each sample type. Since these are non-specific the DNA extraction would be random, minimizing bias. The extensive sequencing coverage achieved (Supplementary Table 3) is sufficient to minimize potential missing data, supporting a lack of sequencing bias. Samples with low sequence coverage were used only for population genetic analyses in which variation in sequencing coverage are explicitly accounted for, or in combination with other samples to generate consensus genome assemblies.

Moreover, we think it unlikely that the sequencing platform used led to bias. The Illumina platform was used for all the sequencing other than for one human *Pneumocystis* isolate, which was sequenced using the MGISEq platform. Studies have shown the 2 platforms to be comparable, including for SNP calling (see (Korostin et al. 2020; Jeon et al. 2019)). For the *P. macacae* assembly, the MinION platform was used as the primary method to produce long reads and scaffolds, while the Illumina sequencing was used to optimize the scaffolds.

How would you explain those samples with low numbers of *Pneumocystis* reads? If we expect that the patients/animals had an active PCP with high numbers of *Pneumocystis* per gram of sample, the samples should have higher numbers of *Pneumocystis* reads. I would just add one or two additional sentences for that.

Response: As noted above a major determinant is the level of infection, which is determined in part by the immune status of the host. Even in hosts with active PCP, the level of infection can be quite variable, as is seen for example in HIV vs. non-HIV patients; a substantial part of the clinical manifestations is related to host immune responses rather than solely to the organism load. The type of sample utilized and the level of host cell contamination, which is highly variable, are also major factors, given our inability to purify the organisms (although we were able to provide some level of enrichment of *Pneumocystis* DNA in more heavily infected lungs). As requested, we have added footnotes to Supplementary Table 2, as follows: “The variability in the percentages of *Pneumocystis* reads among samples depends on a variety of factors, including the type of sample, the level of infection in an individual host, and the DNA enrichment and extraction methods utilized.”. In addition, in this table we have highlighted the samples that were enriched for *Pneumocystis* before sequencing.

All animals, despite three rabbits, were laboratory animals, where the (clinically proven?) PCP was induced by immunosuppression of the animals. Following questions to answer to this topic:
A) Please, state whether the dogs were laboratory animals or not.

Response: Many of the animals had clinical disease, but we do not know the clinical status of the immunocompetent weaning rabbits, which are naturally infected but clear

infection spontaneously. Both dogs were pets, not laboratory animals, as stated in page 24 lines 569-573.

B) The authors mentioned that they tested all samples with mtLSU rRNA PCR and sanger sequencing prior high throughput sequencing. Was it just for confirmation of *Pneumocystis* in the samples? Or did the authors also quantify the *Pneumocystis* organism numbers (equivalents) in the samples with qPCR methods? Even if the copy numbers of e.g. mtLSU gene(s) are different in the animal-derived *Pneumocystis*, it would be helpful to assess the amount of *Pneumocystis* DNA in the samples. Due to the normalization of samples during processing for high throughput sequencing, one cannot quantify the amount of *Pneumocystis* in a DNA extract from a specific sample size. But, I guess, PCR quantification of *Pneumocystis* present in the samples, might help to interpret the relatively low amounts of *Pneumocystis* reads in some samples (see Suppl. Table 2, e.g. *P. jirovecii* Pj46 and Pj54c or *P. macacae* CJ36, etc.). Possibly, the patients/animals did not suffer from an active PCP, but were colonized?

Response: Yes, the mtSLU rRNA PCR and Sanger sequencing were performed only to confirm the identity of *Pneumocystis* in the samples. We attempted to use qPCR to quantify the *Pneumocystis* organism loads in some samples and found that the qPCR results often did not correlate well with the results of Illumina Next Generation Sequencing data NGS (unpublished observation). Consequently, prior to a large-scale genome sequencing we usually conduct a small scale NGS run to quantify the fraction of *Pneumocystis* DNA. This has been clarified in page 26 line 626: “No quantitative PCR methods were used.”. The presence of ~0.1-1% *Pneumocystis* in lung/BAL samples (including the 3 samples pointed out by the reviewer) is typical for patients/animals with active PCP based on our experience; processing to remove contaminating host DNA can enrich for *Pneumocystis* DNA, but is only feasible in a heavily infected sample with a large amount of starting material. As noted above, all animals other than the weaning rabbit had clinical disease (not colonization), but the latter had a yield similar to the other rabbits.

C) Possibly, one of the DNA extraction kits had a lower extraction efficiency either for *Pneumocystis* or for whole DNA than the other kits used? This might also explain some of the samples with low *Pneumocystis* reads. Please explain.

Response: In general, different DNA extraction methods including commercial kits may produce different yields of total DNA but has no significant effect in the ratio of *Pneumocystis* DNA over host DNA unless an enrichment procedure is integrated to selectively increase the *Pneumocystis* DNA. Without enrichment, the DNA extracts normally contain <1% of *Pneumocystis* DNA as discussed above. Although we attempted to enrich *Pneumocystis* DNA before NGS whenever possible, our enrichment protocol requires a large sample volume free of fixative (e.g. formalin in FFPE samples), preventing its use for some types of samples including BALs and FFPE sections.

Additional comments regarding text and graphics:

- Page 23, line 509: change "...obtained France..." to "...obtained from France..."

Response: Corrected. Thank you

- Page 23, line 513: change "...approved for only for research..." to "...approved only for research..."

Response: Corrected.

- Page 37, line 813: write "humans" instead of "Humans"

Response: Corrected.

- Page 42, Fig. 3: the font on the right side is too small to read. Possibly it is better to switch the complete Figure to landscape

Response: The fonts were increased and for sake of clarity KEGG pathways in which the enzymes are involved were replaced by their enzymatic classes. The legend has been updated accordingly.

- Supplementary material, page 55, line 1102: the font size changes in the last word "using"

Response: Corrected.

- Supplementary material, page 56, line 1104, 1106, and 1110: the font size changes in the words/phrases "Retrieved reads...", and two times the word "using"

Response: Corrected.

- Supplementary material, page 62, line 1258: there is a doubled period

Response: Corrected.

- Supplementary material, page 64, line 1286: correct "metablism" to "metabolism"

Response: Corrected.

- Supplementary material, page 66, Figure S1: the font size in the figure is way too small for reading, possibly switch the graphs to landscape and enlarge the font

Response: The fonts were enlarged, and for sake of clarity, the temporal units (x-axis) were changed from "Epoch/Age" to "Period". The computer code to generate this figure is provided at https://github.com/ocisse/pneumocystis_evolution/tree/master/docs/FigS1

- Supplementary material, page 69, legend of Figure S3: “Fixation” should be written uncapitalized

Response: Corrected.

- Supplementary material, page 74, Figure S5: Figures a, b -, and possibly also d are too small for reading the legends / the text, which might be important for understanding the graphs

Response: Corrected.

Reviewer #2 (Remarks to the Author):

In this manuscript the authors sequenced the genome of *Pneumocystis* infecting macaques, rabbits, dogs and rats and compared these genomes to those of species infecting humans, mice and rats. The authors inferred a phylogeny from orthologous genes, identified important genetic features for host adaptation and estimated speciation times relative to the rise of their mammalian hosts.

This article is technically sound and contributes to our understanding of *Pneumocystis* adaptation and evolution. Clearly, it is a relevant research manuscript.

As mentioned by the authors, there are two unexpected finding regarding the phylogeny shown in Figure 2. One of them is the association between *P. wakefieldiae* and *P. murina* and the other is the association between *P. oryctolagi* and the common ancestor of *P. jirovecii* and *P. macacae*.

The evidence supporting the proximity between *P. wakefieldiae* and *P. murina* is quite convincing (the bootstraps, the posterior probabilities, the percentage of trees supporting this association, the genome structure and the mitochondrial phylogeny). And the explanation given to the placement of *P. oryctolagi* (host switching) is perhaps the most parsimonious one. If the authors would like to strengthen further their conclusions they could make a parametric bootstrap analysis to rule out that the unexpected phylogeny is due to long branch attraction (LBA). However, I don't think this analysis is mandatory by any means (i.e. by looking at the tree from figure 2 there seems to be no LBA). Anyway, there is a nice explanation of the parametric bootstrap in: Baum & Smith "Tree Thinking: an introduction to phylogenetic biology" (2013) if the authors think this analysis could complement their results.

Response: We thanks the reviewer for the insightful comment. As noted by the reviewer, there is no evidence suggesting LBA in the phylogeny. The same tree topology was inferred from 106 nucleus-encoded proteins with two maximum likelihood methods (RAxML, IQ-Tree), 568 nuclear protein-coding genes using the Bayesian method BEAST2, 1,718 nucleus-encoded genes using the phylogenetic network tool PhyloNet and 33 mitogenomes using IQ-Tree. Our phylogenetic placement of *Pneumocystis* as sister species of fission yeasts is consistent with a published phylogenies (Fig 1 from (Liu et al. 2009)). LBA is known to plague fission yeasts mitochondrial data because of an

excess of fast evolving sites (Liu et al. 2009), although no evidence of a similar phenomenon in *Pneumocystis* has been found. Given the reasons stated above, we feel that additional parametric bootstrapping analyses as implemented more recently ((Goldman 1993; Huelsenbeck 1996) are unnecessary.

We agree that our explanation of *P. oryctolagi* placement is the most parsimonious. Improved taxon sampling in the clade A (Fig 2) in the future will potentially help clarify its position in the future.

Regarding the phylodating, it is not clear for me why the authors used as a calibration prior a median time of 65 mya for the divergence between *P. jirovecii*/*P. macacae* since the divergence between human and macaque is estimated to be approximately 20 mya. Further explanation in the manuscript would make the use of this prior more clear for the readers. In addition, it is important that the authors evaluate the impact of the priors in their results by running the analyses without the sequences (i.e. sampling from priors) and to test whether sequences are informative on estimated divergence dates (see Drummond & Bouckaert "Bayesian Evolutionary Analysis with Beast" (2015)).

Response: The 65 Mya prior for *P. jirovecii*/*P. macacae* separation is from our previous publication (Cisse et al. 2018), which was estimated using two secondary calibrations: a separation of fission yeasts (*Schizosaccharomyces*) and *Pneumocystis* at ~ 467 mya ((Beimforde et al. 2014); this reference is based on combined fossil and molecular data) and 100 mya as a minimum age for the *Pneumocystis* genus ((Keely, Fischer, and Stringer 2003); – this reference is based on independent molecular data). Of note while setting prior is necessary in any Bayesian analysis, there is no general consensus for how priors should be set in all circumstances. Our use of informed priors follows the recommended practice in Bayesian phylogenetic analyses (Nelson, Andersen, and Brown 2015).

To evaluate the impact of the priors in our results, we sampled from various priors and found no significant difference between the marginal posterior distribution on rate and the marginal prior distribution (data not shown but available upon request). This indicates that the posterior simply reflects the prior (<https://www.beast2.org/tree-priors-and-dating/>). To clarify, we have added the following statement at the page 32 lines 774 - 777: “We sampled from various priors and found no significant difference between the marginal posterior distribution on rate and the marginal prior distribution. This indicates that the posterior reflects the prior”.

To evaluate the whether the sequences are informative on estimated divergence dates, we used BEAST2 and 24 single-copy nucleus encoded genes shared by all *Pneumocystis* species and *S. pombe* for computational efficiency (dataset 1 -- Material and Methods page 31 line 767). Different values for the *P. jirovecii*/*P. macacae* prior ranging from 20 to 80 mya implemented as soft bound distributions (log normal) were evaluated. Of note, soft bound distributions are often superior to hard bound distributions (e.g. uniform) for measuring uncertainty. Also, we tested 20 mya as prior because it roughly corresponds to the human-macaque divergence. Applying recent priors caused fatal numerical errors and further investigation revealed that a prior of 20 mya is incompatible with the sequence data. Subsequently, all prior values younger than ~50 mya failed while priors with a minimum ~55 Mya succeed with evolutionary ages

consistent with our reported values. These results suggest that sequences are informative on estimated ages.

Minor comments

Give more details on how the Shimodaira-Hasewaga test was used. The readers would appreciate this.

Response: Complete details have been added to the text at page 31 lines 743-750.

In line 394 the authors write that "msg genes may have a polyphyletic origin". Although the authors clarify this by later writing that "i.e., distinct families were present in the most recent ancestors of *Pneumocystis*" this may be a bit confusing. Do the authors mean that msg really originated independently or that they originated once and diversified before the last common ancestor of extant *Pneumocystis* species?

Response: We meant that we cannot definitely resolve the origin of the msg genes because a convergent evolution cannot be ruled out. There are two non-necessarily competing hypotheses here: (i) msg families have a single origin and then diverged rapidly in different species or (ii) msg genes emerged independently in different species as a response to selective pressure e.g. host immune systems (convergent evolution). The high level of recombination and substitution rates in msg genes render a precise delineation of the two hypotheses impractical. However, gene gain/loss as well as evolutionary ages of the distinct families suggest a stepwise emergence of msg genes during *Pneumocystis* evolution. For instance, msg-type E and B are ancient and would fit the first hypothesis whereas C and D as well as A subfamilies are more recent (Figure 6). An intriguing feature is that A subfamilies are clearly related but we found no evidence that they are derived from each other. Instead ancestral reconstruction suggests that they appeared at roughly similar times or at least have co-existed for a long time. Again, it's possible that rapid evolution and/or extensive recombination make difficult to precisely delineate the origin of these families.

We have clarified the issue at page 19 line 438: "although convergent evolution of msg cannot be ruled out".

Lines 481 and 482. The authors used the word "further" three times in a single sentence.

Response: Corrected. Thanks

Luis Delaye

Reviewer #3 (Remarks to the Author):

This study is impressive and provides new data on the evolution of *Pneumocystis* and its adaptation to hosts.

- L63 : this is not fully true in rodents since different rodent species can be infected by the same *Pneumocystis* species. PMID: 29117878 ; PMID: 31247340. Corresponds to ref 10, 11. Please rephrase

Response: There is no consensus on this subject yet. The two studies mentioned by the reviewer suggests a possible relaxation in host specificity among rodents. We believe that their findings represent an interesting paradigm-shifting hypothesis that still awaits validation. The reason is that the evidences presented in both studies are all based on analysis of only a few short genetic loci (mtLSU and mtSSU) (nested PCR reactions in the case of ref 10). In the absence of at least whole genome sequencing data, the true nature of the so-called *Pneumocystis* species cannot be asserted. This is important because it is impossible to differentiate, based on few small genomic regions, whether these *Pneumocystis* organisms represent bona-fide species, different strains, recombinants, or divergent haplotypes. The hypothesis of strong host specificity in all formally named *Pneumocystis* species is generally accepted because it's a generalization from failures of experimental cross infections in SCID mice and rats (refs 1 and 2). Our statement refers to species that have been clearly delineated by accepted taxonomic methods and supported by experimental validations. To clarify, we've rephrased as follows at page 4 lines 60-61: "Cross-species inoculation studies with *P. jirovecii* and *P. carinii* have found that they can infect only humans and rats, respectively.".

- Line 148-150: no fully true regarding ref 10 and 11. Other hypothesis?

Response: This refers to the following statement: "There are clearly fewer rearrangements among rodent *Pneumocystis* species (*P. wakefieldiae*, *P. carinii* and *P. murina*) than among all other species (Fig. 1; Supplementary Table 4), which is likely due to their younger evolutionary ages and closer taxonomic relationships of their host species". For sake of clarity, we have removed the underlined section from the sentence at page 8 line 159.

Chromosomal rearrangements (CRs) can create inversions, which act as recombination suppressors. Recombination suppression leads to the differentiation of rearranged genes, which in turn accelerates the accumulation of genetic difference among populations. When fixed, these mutations cause incompatibilities and pose important reproductive barriers between species (Rieseberg 2001). If CRs participate in *Pneumocystis* speciation, one can argue that the fewer rearrangements among rodents infecting *Pneumocystis* reflect the high similarities in cellular architectures among rodents, that is, fewer CRs are needed compared to other *Pneumocystis*. However, CRs are complex events, which can be caused or influenced by many additional factors such as the frequency of recombination, the strength and direction of selection, presence of gene flow or active repetitive elements. In addition, CRs are ancient and accumulated slowly (roughly 1.3 events per mya in this case), so the original cause(s) of rearrangements are likely not relevant anymore. This is important because most CRs are strongly deleterious and only retained under strong selection. Given the many uncertainties, we have refrained from speculating on this in the manuscript.

- Can high variability of *P. macacae* among individuals be explained by the use of Nanopore which is known to introduce technical errors. Illumina reads should be able to correct some of these errors. More details about the animals should be interesting to provide to understand the diversity obtained.

Response: We feel that a technical variability due to Nanopore sequencing is unlikely, since as the reviewer suggests, Illumina reads were used to correct Nanopore reads (this is mentioned in Material and Methods at page 27 lines 652-654: “*Pneumocystis* nanopore reads were assembled using Canu v.1.8.0 (49), overlapped with the assembly using Racon v.1.3.3 (50) and polished with Pilon v1.22 (51) using the Illumina reads aligned with BWA MEM v0.7.17 (52)”). The resulting assembly was manually inspected for inconsistencies.

A biological variability is difficult to assess because all macaques were maintained in captivity, experimentally infected with the simian immunodeficiency virus (SIV) and presented variable clinical symptoms. While a weakened host immunological status is critical for PCP colonization, how it directly influences the genetic diversity of *P. macacae* is difficult to predict.

Only consensus sequence has been used? Only one of the 6 sequences?

Response: The *P. macacae* genome assembly is not a consensus sequence and has been built exclusively using samples from one infected macaque. While all six samples were sequenced, only one was used for de novo genome assembly because of its high *Pneumocystis* DNA content relative to host contamination (68%; sample P2C, Table 2). The *P. macacae* reference genome assembly was generated using Nanopore and Illumina reads from DNA samples from a single macaque with heavy infection (P2C, Table 2). Sequences from other *P. macacae* samples were not used for genome assembly of the reference genome but exclusively used for population genetic analyses. This has been noted in the manuscript, page 26 lines 640-642: “Nanopore sequencing was performed on *P. macacae* DNA samples prepared from a single heavily infected macaque (P2C) with ~68% *Pneumocystis* DNA based on prior Illumina sequencing”.

- It would be important (does not exist in the literature) to compare interindividual variation as compared to interspecies variation. Indeed, the authors found a high (12%) variation % in 6 macaques which can be corresponding to different species as well. Please add an analysis regarding this to understand how individual *Pneumocystis* varies inside the species as compared to the inter species variation.

Response: The high level of SNP polymorphisms (12%) in *P. macacae* is caused by highly polymorphic regions (especially the *Msg* gene family). A high level of diversity is also seen in human *Pneumocystis* (Cisse et al. 2018; Ma et al. 2016) and again is primarily related to diversity in the *Msg* superfamily. The effect disappears when these regions are excluded from the analysis. We have modified the sentence at page 6 lines 125 - 126 (modified text is underlined): “Post assembly mapping revealed a ~~non-~~

negligible amount of genetic variability among samples, for example the average genome wide single nucleotide polymorphism (SNP) diversity among six *P. macacae* isolates excluding highly polymorphic regions such as Msg genes is only ~ 0.1% .”.

The supplementary analysis requested by the reviewer is already covered by population genomics analyses (F_{ST}), in which we have compared genetic data from *P. jirovecii*, *P. macacae* and *P. oryctolagi* samples (Supplementary Figure 3 and discussed in the text at the page 11 lines 233-251. F_{ST} compares the genetic variability within and between population. The values range from 0 to 1, where 0 indicates a complete panmixis and 1 a complete separation without gene flow. In this study, we found a significant interspecies divergence between *P. jirovecii*, *P. macacae* and *P. oryctolagi* populations ($F_{ST} > 0.8$).

The question is also partially answered by the pairwise genome difference calculated from alignments of complete or draft genome sequences (Supplementary Table 5). Only samples for which the original authors provided a draft genome were initially included. Pairwise divergence among three *P. jirovecii* assemblies already showed a mean of 0.3% whereas all three assemblies display > 15% of genomic divergence when compared with other *Pneumocystis* species (Supplementary Table 5). At the reviewer request, we have now included an additional four *P. jirovecii* and five *P. macacae* low coverage samples. Since the low sequencing depth does not allow for *de novo* assembly, we generated genome sequences by identifying genetic variants relative to reference genomes and by integrating those differences into a consensus sequence. Our results show that interspecies divergence significantly exceeds intra species divergence, which is consistent with a complete species separation. We have reported the numerical results in page 8 lines 165 – 176: “To understand the relationship between the intra-species and inter-species genetic diversity of *Pneumocystis*, we used a mapping approach to generate additional genome assemblies for four *P. jirovecii* and five *P. macacae* samples with low sequence coverage (see Material and Methods). All data are expressed as the mean \pm standard deviation. The pairwise intra species genome divergences among *P. jirovecii* genome assemblies ($0.3 \pm 0.2\%$, $n = 8$) are significantly lower than those obtained when comparing them to *P. macacae* assemblies ($16.1 \pm 0.2\%$, $n = 5$) (two sample t-test, p -value = 1.4×10^{-14}) or to other *Pneumocystis* species ($21.6 \pm 0.9\%$, $n = 7$) ($p = 2.9 \times 10^{-10}$). Similarly, mean divergence among *P. macacae* genome assemblies ($0.8 \pm 0.3\%$, $n = 5$) is lower than divergence when they are compared to *P. jirovecii* assemblies ($15.6 \pm 0.2\%$, $n = 8$) ($p = 1.3 \times 10^{-10}$) or other *Pneumocystis* species genome assemblies ($21.8 \pm 0.9\%$) ($p = 7.2 \times 10^{-11}$).”. In addition, we’ve updated the material and methods at pages 29-30 lines 704-711) and added a footnote to the Supplementary table 5.

Literature Cited:

- Beimforde, C., K. Feldberg, S. Nylinder, J. Rikkinen, H. Tuovila, H. Dorfelt, M. Gube, D. J. Jackson, J. Reitner, L. J. Seyfullah, and A. R. Schmidt. 2014. 'Estimating the Phanerozoic history of the Ascomycota lineages: combining fossil and molecular data', *Mol Phylogenet Evol*, 78: 386-98.
- Cisse, O. H., L. Ma, D. Wei Huang, P. P. Khil, J. P. Dekker, G. Kutty, L. Bishop, Y. Liu, X. Deng, P. M. Hauser, M. Pagni, V. Hirsch, R. A. Lempicki, J. E. Stajich, C. A. Cuomo, and J. A. Kovacs. 2018. 'Comparative Population Genomics Analysis of the Mammalian Fungal Pathogen *Pneumocystis*', *MBio*, 9: e00381-18.
- Goldman, N. 1993. 'Statistical tests of models of DNA substitution', *J Mol Evol*, 36: 182-98.
- Huelsenbeck, JP., Hillis DM, Nielsen Rasmus. 1996. 'A Likelihood-Ratio Test of Monophyly', *Systematic Biology*, 45: 546-58.
- Jeon, S. A., J. L. Park, J. H. Kim, J. H. Kim, Y. S. Kim, J. C. Kim, and S. Y. Kim. 2019. 'Comparison of the MGISEQ-2000 and Illumina HiSeq 4000 sequencing platforms for RNA sequencing', *Genomics Inform*, 17: e32.
- Keely, S. P., J. M. Fischer, and J. R. Stringer. 2003. 'Evolution and speciation of *Pneumocystis*', *J Eukaryot Microbiol*, 50 Suppl: 624-6.
- Korostin, D., N. Kulemin, V. Naumov, V. Belova, D. Kwon, and A. Gorbachev. 2020. 'Comparative analysis of novel MGISEQ-2000 sequencing platform vs Illumina HiSeq 2500 for whole-genome sequencing', *PLoS One*, 15: e0230301.
- Liu, Y., J. W. Leigh, H. Brinkmann, M. T. Cushion, N. Rodriguez-Ezpeleta, H. Philippe, and B. F. Lang. 2009. 'Phylogenomic analyses support the monophyly of Taphrinomycotina, including *Schizosaccharomyces fission yeasts*', *Mol Biol Evol*, 26: 27-34.
- Ma, L., Z. Chen, W. Huang da, G. Kutty, M. Ishihara, H. Wang, A. Abouelleil, L. Bishop, E. Davey, R. Deng, X. Deng, L. Fan, G. Fantoni, M. Fitzgerald, E. Gogineni, J. M. Goldberg, G. Handley, X. Hu, C. Huber, X. Jiao, K. Jones, J. Z. Levin, Y. Liu, P. Macdonald, A. Melnikov, C. Raley, M. Sassi, B. T. Sherman, X. Song, S. Sykes, B. Tran, L. Walsh, Y. Xia, J. Yang, S. Young, Q. Zeng, X. Zheng, R. Stephens, C. Nusbaum, B. W. Birren, P. Azadi, R. A. Lempicki, C. A. Cuomo, and J. A. Kovacs. 2016. 'Genome analysis of three *Pneumocystis* species reveals adaptation mechanisms to life exclusively in mammalian hosts', *Nat Commun*, 7: 10740.
- Nelson, B. J., J. J. Andersen, and J. M. Brown. 2015. 'Deflating trees: improving Bayesian branch-length estimates using informed priors', *Syst Biol*, 64: 441-7.
- Rieseberg, L. H. 2001. 'Chromosomal rearrangements and speciation', *Trends Ecol Evol*, 16: 351-58.

REVIEWERS' COMMENTS:

Reviewer #1 (Remarks to the Author):

The manuscript by Cissé et al. entitled "Genomic insights into the host specific adaptation of the *Pneumocystis* genus and emergence of the human pathogen *Pneumocystis jirovecii*" is very well written and clear. The methods used are comprehensive and included analysis of the genomic, as well as mitochondrial genes, transcriptome analysis, and phylodating.

Comprehensive research regarding *Pneumocystis* species is still sparse. This article allows a deep insight in the genetics but also possible biochemical processes in five *Pneumocystis* species (*P. jirovecii*, *P. macacae*, *P. oryctolagi*, *P. carinii*, and *P. wakefieldiae*) from different mammalian hosts. Within this article, the complete genomes of the four animal-derived species were analysed for the first time and were compared to the three previously published genomes of *P. jirovecii* strains, *P. carinii*, and *P. murina*.

This outstanding manuscript allows a closer insight in the complex genetic rearrangements done by *Pneumocystis* for adaptation to the host species, which is implicitly necessary for understanding *Pneumocystis* spp. and *Pneumocystis* pneumonia.

All comments mentioned in the first review were answered clearly. Thank you very much!

With best regards

Reviewer #2 (Remarks to the Author):

I consider the authors have addressed all my queries satisfactorily. This is a nice article that will advance our understanding of *Pneumocystis* evolution.

Reviewer #3 (Remarks to the Author):

No specific comment.